# Loss of cell-autonomously secreted laminin-α2 drives muscle stem cell dysfunction in LAMA2-related muscular dystrophy

Timothy J. McGowan [1], Judith R. Reinhard [1], Nicolas Lewerenz [1], Marta Białobrzeska[2], Shuo Lin[1], Jacek Stępniewski[2], Krzysztof Szade [3], Józef Dulak [2] & Markus A. Rüegg [1] ✉

The extracellular matrix protein laminin-α2 is essential for preserving the integrity of skeletal muscle fibers during contraction. Its importance is reflected by the severe, congenital LAMA2-related muscular dystrophy (LAMA2 MD) caused by loss-of-function mutations in the *LAMA2* gene. While laminin-α2 has an established role in structurally supporting muscle fibers, it remains unclear whether it exerts additional functions that contribute to the maintenance of skeletal muscle integrity. Here, we report that in healthy muscle, activated muscle stem cells (MuSCs) express *Lama2* and remodel their microenvironment with laminin-α2. By characterizing LAMA2 MD-afflicted MuSCs and generating MuSC-specific *Lama2* knockouts, we show that MuSC-derived laminin-α2 is essential for rapid MuSC expansion and regeneration. In humans, we identify *LAMA2* expression in MuSCs and demonstrate that loss-of-function mutations impair cell-cycle progression of myogenic precursors. In summary, we show that self-secreted laminin-α2 supports MuSC proliferation post-injury, thus implicating MuSC dysfunction in LAMA2 MD pathology.

LAMA2-related muscular dystrophy (LAMA2 MD or MDC1A) is a devastating congenital muscular dystrophy caused by mutations in the *LAMA2* gene. In the most severe cases, these mutations result in an absence of laminin-α2[1], an extracellular matrix (ECM) protein that forms heterotrimeric complexes with laminin β and γ chains[2]. In skeletal muscle, laminin-α2 is assembled with laminin-β1/β2 and laminin-γ1 to form laminin-211 and laminin-221[3]; however, for simplicity, we will refer only to laminin-α2. These laminin-α2-containing complexes are the principal constituents of the muscular basement membrane (BM)[4], an assembly of ECM proteins that maintains tissue integrity by surrounding and structurally supporting muscle fibers. Laminin-α2's importance is underscored by the severe muscle fiber degeneration, inflammation and fibrotic depositions commonly observed in muscle biopsies from LAMA2 MD patients[5]. Clinically, the absence of laminin-α2 manifests through neonatal hypotonia, muscle weakness, ambulatory failure and respiratory complications for which no curative treatments are available[5–7].

Due to laminin-α2's established role as a structural support protein, the dystrophic pathology of LAMA2 MD has thus far primarily been attributed to muscle fiber frailty. The recent identification of aberrant changes in the endothelial cell compartment of a LAMA2 MD mouse model[8] revealed that multiple skeletal muscle components are impacted by the loss of laminin-α2, suggesting that the protein may have undescribed functions within the tissue. There are hints that some of laminin-α2's additional functions involve muscle stem/satellite cells (MuSCs), as regenerative defects have been observed on multiple occasions in mouse models of LAMA2 MD[8–10].

MuSCs are somatic stem cells that reside between muscle fibers and the BM[11]. Typically resting in a state of quiescence, MuSCs respond to muscle damage by activating and proliferating to produce a pool of

[1]Biozentrum, University of Basel, Basel, Switzerland. [2]Department of Medical Biotechnology, Faculty of Biochemistry, Biophysics and Biotechnology, Jagiellonian University, Kraków, Poland. [3]Laboratory of Stem Cell Biology, Faculty of Biochemistry, Biophysics and Biotechnology, Jagiellonian University, Kraków, Poland. ✉e-mail: markus-a.ruegg@unibas.ch

progenitor cells. They can then self-renew and return to quiescence to maintain a pool of stem cells, or differentiate to regenerate muscle fibers[11–13]. Through their regenerative function, MuSCs can replace degenerated fibers and slow disease progression in muscle-wasting diseases[8,14]. However, it was revealed that in certain muscular dystrophies, fiber degeneration outpaces fiber regeneration due to cell-extrinsic and/or cell-intrinsic impairments of MuSC function[15,16].

Cell-extrinsic disruptions of MuSC function occur when the dystrophic milieu interferes with the cells' regenerative capacities. There are suggestions that this may occur in LAMA2 MD mouse models due to alterations of the BM stiffness[17] and composition[18], increased intramuscular fibrosis[19–21] and high levels of inflammation[22,23]. Moreover, targeting vascular defects in LAMA2 MD mice was shown to increase MuSC abundance[8].

MuSC function can also become compromised independently of the microenvironment in muscular dystrophies where the disease-causing mutation intrinsically disrupts the cells' regenerative properties. In Duchenne Muscular Dystrophy (DMD), for example, MuSCs suffer from an intrinsic polarity defect caused by the loss of dystrophin expression. Consequently, MuSCs cannot generate a pool of myogenic progenitor cells, and muscle repair is hindered in mouse models of the disease[24]. The evidence that cell-intrinsic factors contribute to an impairment of MuSC function and thereby exacerbate dystrophic pathology has generated novel therapeutic opportunities to combat muscle wasting[14,25]. For example, in DMD, targeting MuSCs and rescuing their polarity defect improved muscle regeneration and increased tissue size and contractile force[25]. These outcomes underscore the importance of identifying MuSC impairments in other muscular dystrophies, as similar therapeutic approaches could be taken to slow disease progression.

Here, we report that in healthy muscle, proliferating MuSCs remodel their microenvironment with laminin-α2. We demonstrate that MuSC-derived laminin-α2 is essential for MuSC expansion, as LAMA2 MD-afflicted MuSCs exhibit proliferative defects that are not rescued by transplantation into non-dystrophic muscle. Furthermore, we show that a MuSC-specific *Lama2* knockout is sufficient to slow MuSC proliferation and delay muscle regeneration. Finally, the same phenotype is observed in *LAMA2*-deficient human myogenic precursor cells derived from induced pluripotent stem cells. Interestingly, transcriptomic analyses of *LAMA2*-deficient myogenic precursor cells reveal the downregulation of genes involved in cell-cycle progression. All in all, our work describes a crucial role for MuSC-derived laminin-α2 in muscle regeneration and reveals that MuSC function is intrinsically impaired in LAMA2 MD.

## Results

### Activated MuSCs express *Lama2* and remodel their micro-environment with laminin-α2

Though MuSCs are anatomically located between muscle fibers and the BM's laminin-α2, it is not known whether they are a source of laminin-α2 in skeletal muscle. Single-nucleus RNA-sequencing (snRNA-seq) of adult murine skeletal muscle[26] identified *Lama2* expression in a multitude of cell types and myonuclei (Fig. S1a). These included myonuclei located at the fibers' neuromuscular and myotendinous junctions, fibro-adipogenic progenitor cells (FAPs) and MuSCs (Fig. S1a). To confirm that MuSCs express *Lama2*, we isolated Integrin-α7 + MuSCs from 8-week-old C57BL/6 mice using fluorescence-activated cell sorting (FACS) and cultured them on collagen-coated tissue culture dishes. As culturing MuSCs reduces their stemness[27], these cells are hereafter referred to as primary myoblasts (PMs). Single-molecule RNA fluorescence in situ hybridization (smRNA FISH) confirmed the presence of *Lama2* mRNA in the *Pax7*-expressing PMs (Fig. 1a). In this experiment, we observed higher *Lama2* abundance in *Pax7*-expressing PMs after their first passage compared to when they were freshly-isolated, indicating that *Lama2* expression increases in

PMs as they activate and proliferate ex vivo (Fig. S1b, S1c). We also observed that *Lama2* abundance was strongly reduced in differentiating cells that had lost *Pax7* expression and upregulated *Myog* expression (Figs. 1a, S1d). In line with this observation, RT-qPCR revealed that PMs cultured in differentiation medium for 24 h significantly downregulated *Lama2* mRNA concomitant with the loss of *Pax7* compared to isogenic cells that were maintained in proliferation medium and continued to express *Pax7* (Fig. 1b). The same pattern was seen at the protein level, with Pax7-positive (Pax7+) cells co-localizing with a stronger laminin-α2 signal than isogenic Myogenin-positive (Myogenin+) cells in the same culture dish (Fig. 1c, d). This indicated that PMs express *Lama2* and produce laminin-α2 as they proliferate ex vivo, before downregulating its expression and production as they differentiate.

To assess whether these findings could be confirmed in vivo, we analyzed single-cell RNA-sequencing (scRNA-seq) datasets generated from injured skeletal muscle[28] and sub-clustered MuSCs into quiescent, activated, differentiating or differentiated cells based on the expression of known myogenic factors and cell-cycle markers (Fig. S1e). Similarly to what we had seen ex vivo (Figs. 1a, b, S1b–S1d), *Lama2* was expressed in quiescent and activated MuSCs, downregulated in differentiating MuSCs and largely not expressed in mature muscle (Fig. 1e). By combining smRNA FISH with immunostaining in uninjured *tibialis anterior* (TA) muscle of 8-week-old C57BL/6 mice, we confirmed that quiescent MuSCs express *Lama2* (Fig. S1f, S1g). We further substantiated the scRNA-seq results by showing that 7 days post-cardiotoxin (CTX) injury, Pax7+ cells were strongly positive for *Lama2* mRNA (Fig. 1f, green arrows), while differentiating, Myogenin+ cells were largely negative (Fig. 1f, orange arrows). Taken together with the ex vivo results (Figs. 1a, b, S1d), these findings show that *Lama2* is expressed in a state-dependent manner by MuSCs. While *Lama1* was not expressed in MuSCs, *Lama4* and *Lama5* had similar expression patterns as *Lama2*, albeit at lower levels (Fig. S1h).

To visualize if activated MuSCs remodeled their microenvironment with laminin-α2 post-injury, we generated a mouse model with tamoxifen-inducible EGFP expression in MuSCs by crossing Pax7[CreER] mice[29] with mice that were engineered to contain a Cre-inducible CAG-EGFP transgene in the *Rosa 26* locus[30]. After 5 consecutive days of tamoxifen treatment, we injected CTX into their TAs and collected the muscles at 2 days post-injury (DPI) for whole-mount staining. At this early time point of regeneration, EGFP+ cells with the characteristic morphology of proliferating MuSCs[31] were surrounded by laminin-α2 (Fig. 1g). Whilst this protein is restricted to the MuSCs' basal side in uninjured conditions[31], we also detected laminin-α2 on their apical side as they proliferated inside the BM remnants (Fig. 1g, orange arrow). We confirmed this localization in cross-sections of the TA at 4 DPI, as laminin-α2 was not only present in the BM remnants of ghost fibers[32] on the basal side of EGFP+ cells, but also on the cells' apical sides within ghost fibers (Fig. 1h, box 2, orange arrows, Supplementary Movie 1). We also observed that EGFP+ cells located in the interstitial space between regenerating fibers co-localized with laminin-α2 (Fig. 1h, box 1, Supplementary Movie 1). Taken together, these observations suggest that some of the laminin-α2 detected in muscle post-injury is newly synthesized by MuSCs. To estimate the amount of newly synthesized laminin-α2, we collected TA muscles from injured and contralateral, uninjured legs at 4 and 10 DPI and measured the relative abundance of laminin-α2 by Western blot analysis. Although variability among samples limited definitive conclusions, each mouse showed a trend towards higher levels of laminin-α2 in the injured TA compared to the uninjured TA at 4 DPI, but not at 10 DPI (Fig. S2a, S2b). This increase in laminin-α2 after injury is consistent with quantifications of *Lama2* mRNA levels at 4 DPI[18] and further supports our data that activated MuSCs strongly express *Lama2* (Fig. 1e). However, other muscle-resident cells, such as FAPs (Fig. S1a), may also secrete laminin-α2. To remove any potential influence from such cells and confirm that

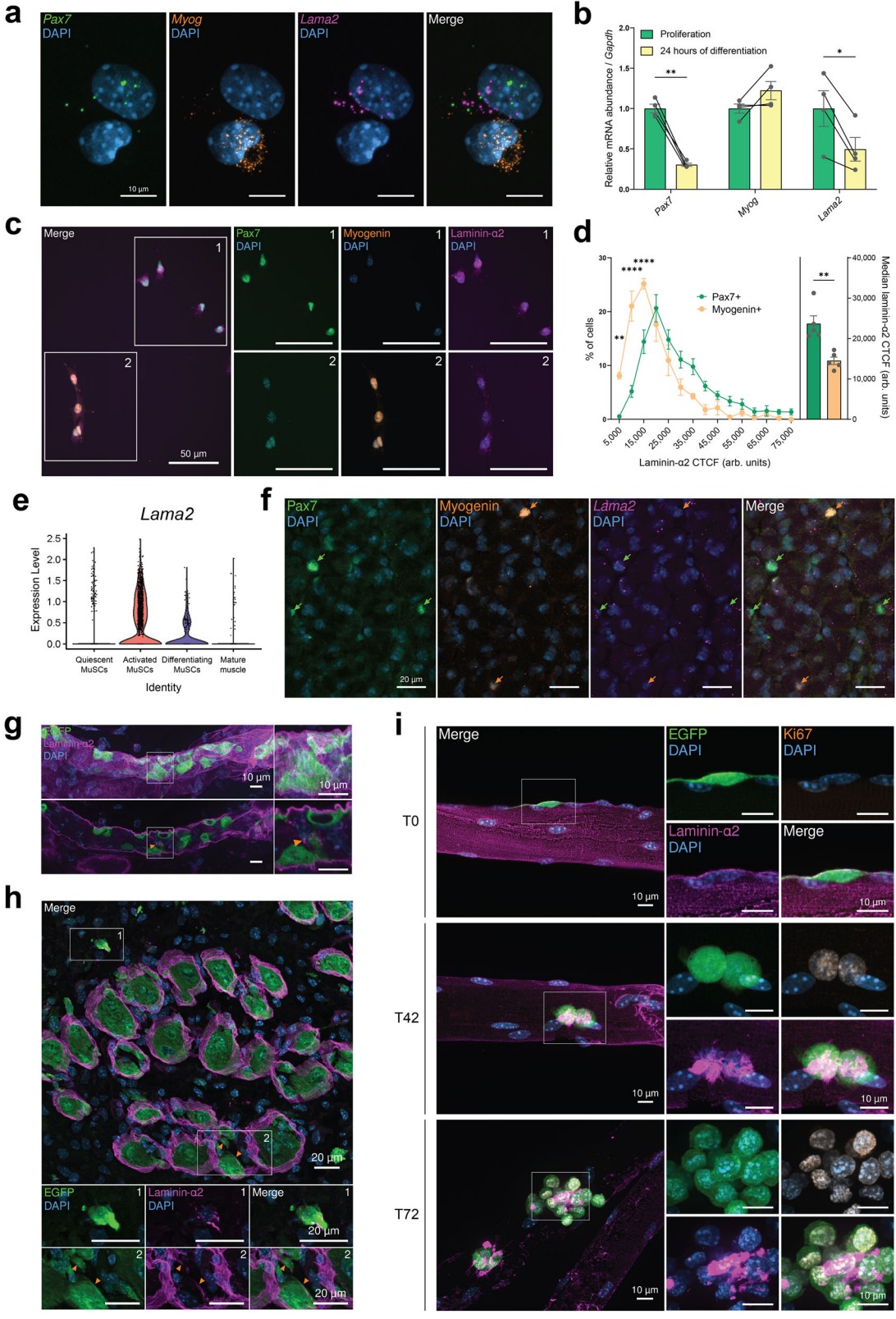

MuSCs secrete laminin-α2, we isolated single fibers from the *extensor digitorum longus* (EDL) muscle of mice with EGFP-labeled MuSCs and maintained them in culture for 0 (T0), 42 (T42) or 72 (T72) h. At T0, the Ki67-negative, quiescent EGFP+ MuSCs were located in their anatomical niche between the laminin-α2-positive BM and the muscle fiber (Fig. 1i). By T42, MuSCs were Ki67-positive (Ki67+) and now co-localized with abundant laminin-α2 signal (Fig. 1i), which could be observed on single fibers from afar (Fig. S2c). At T72, the cells continued to proliferate and formed clusters in which laminin-α2 could be detected (Fig. 1i), confirming that activated MuSCs secrete laminin-α2 into their microenvironment. We also tested whether MuSC differentiation on EDL fibers led to a reduction in co-localization with laminin-α2 by staining EGFP+ cells for Myogenin and laminin-α2. No clear co-localization patterns were observed, as some differentiating

**Fig. 1 | Activated MuSCs express _Lama2_ and remodel their microenvironment with laminin-α2. a** smRNA FISH performed on PMs in cell culture (_Pax7_ in green, _Myog_ in orange, _Lama2_ in magenta, DAPI in blue). **b** Quantification of _Pax7_, _Myog_ and _Lama2_ mRNA levels by RT-qPCR. Lines link PMs isolated from the same mouse cultured in proliferation medium and after 24 h in differentiation medium. _n_ = 4 mice. **c** Representative laminin-α2 immunostaining in PM culture (Pax7 in green, Myogenin in orange, laminin-α2 in magenta, DAPI in blue). **d** Distribution (left) and median (right) laminin-α2 Corrected Total Cell Fluorescence (CTCF) in Pax7+ and Myogenin+ cells. _n_ = 5 mice. **e** Violin plot of _Lama2_ expression from a scRNA-seq dataset[28]. Markers used to sub-cluster MuSCs are shown in Fig. S1e. **f** Representative immunostaining and smRNA FISH in TA muscle 7 DPI (Pax7 in green, Myogenin in orange, _Lama2_ in magenta, DAPI in blue). Green arrows: Pax7+ cells; orange arrows: Myogenin+ cells. **g** Whole-mount immunostaining of EGFP-labeled MuSCs in TA

muscle 2 DPI (EGFP in green, laminin-α2 in magenta, DAPI in blue). Upper panel: maximal intensity projection; lower panel: one focal plane inside the ghost fiber; orange arrow: laminin-α2 presence in the ghost fiber. **h** Cross-sectional immunostaining of TA muscle 4 DPI (EGFP in green, laminin-α2 in magenta, DAPI in blue). Box 1: interstitial EGFP+ cell positive for laminin-α2; box 2: orange arrows highlight laminin-α2 presence on the apical sides of EGFP+ cells in ghost fibers. **i** Immunostaining of EDL fibers after 0 (T0), 42 (T42) and 72 (T72) h in culture (EGFP in green, Ki67 in orange, laminin-α2 in magenta, DAPI in blue). In **f**–**i**, experiments were performed once (**f**–**h**) or twice (**i**) with _n_ = 1 (**f** and **g**), _n_ = 4 (**i**) or _n_ = 5 (**h**) mice. Repeated experiments showed similar results. Data are means ± SEM. Statistical significance was determined by paired Student's two-sided _t_-test (**b**) or two-way ANOVA with Bonferroni's multiple comparisons test (**d**). **_P_ < 0.01; ***_P_ < 0.001. Source data and _P_-values are provided as a Source Data file.

Myogenin+/EGFP+ cells co-localized with laminin-α2 while others did not (Fig. S2d). Since laminin-α2 is mostly present in skeletal muscle as laminin-211[3], a hetero-trimeric complex that contains laminin-β1 and γ1 chains, we examined whether MuSCs also express _Lamb1_ and _Lamc1_, the genes encoding these respective chains. Analyses of scRNA-seq datasets[28] revealed that activated MuSCs express _Lamb1_ and _Lamc1_ (Fig. S2e). To determine if the cells secrete laminin-211, we stained cultured EDL fibers at T0, T42 and T72 with anti-laminin-α2 and anti-laminin-β1-γ1 antibodies. The proteins co-localized at all time points studied, and increases in laminin-α2 were concomitant with increases in laminin-β1-γ1 (Fig. S2f).

## Muscle regeneration is impaired in _dy^W_/_dy^W_ mice

The finding that activated MuSCs remodel their microenvironment with laminin-α2 suggested that the failure of MuSCs to secrete laminin-α2 might affect regeneration in _Lama2_-deficient mice. To investigate this, we next compared the levels of muscle repair in _dy^W_/_dy^W_ and _dy^3K_/_dy^3K_ mice, the two most used LAMA2 MD mouse models, and their wild-type littermates at 4 weeks of age. Immunostainings revealed a high proportion of embryonic myosin heavy chain-positive fibers in the TAs of both LAMA2 MD mouse models (Fig. S3a, S3b), indicating ongoing muscle degeneration and regeneration. In agreement with this, LAMA2 MD mice also had a higher proportion of activated Ki67+/Pax7+ cells (Fig. S3c) and an increased number of Pax7+ and Myogenin+ cells within the _triceps_ (TRC) and TA muscle compared to wild-type mice (Fig. S3d, S3e). To confirm that MuSCs were activated and progressing through the cellcycle, we injected wild-type and _dy^W_/_dy^W_ mice with the nucleotide analog EdU. Twenty-four hours later, only _dy^W_/_dy^W_ MuSCs had incorporated EdU (Fig. S3f, S3g), indicating that they had progressed through S phase while wild-type MuSCs had not. Collectively, these data demonstrate that at 4 weeks of age, MuSCs are highly activated due to ongoing cycles of muscle fiber degeneration and regeneration in LAMA2 MD mice, while wild-type MuSCs are largely quiescent.

To directly compare MuSC function, we maximized MuSC activation by injecting the TAs of 4-week-old wild-type and _dy^W_/_dy^W_ mice with CTX and collecting the muscles at 4, 7 and 14 DPI for analyses (Fig. 2a). For each mouse, the contralateral TA was also harvested and used as an uninjured control for normalizations of muscle mass and fiber numbers, as both are strongly reduced in _dy^W_/_dy^W_ mice[4]. Comparisons of TA masses revealed that at 4 DPI, injured TAs of _dy^W_/_dy^W_ mice exhibited significantly lower relative muscle mass compared to those of wild-type mice (Fig. 2b). Though they had a similar relative number of fibers per cross-section at this early time point (Fig. 2c), the regenerating muscles of _dy^W_/_dy^W_ mice contained regions that were occupied by interstitial cells and a large interstitial space separating the regenerating fibers (Fig. 2d, orange arrows). Muscle mass subsequently increased in both mice, but at different rates: by 14 DPI, injured TAs were on average 17% lighter than their contralateral uninjured TAs in _dy^W_/_dy^W_ mice, while in wild-type mice, injured TAs were now 9% heavier than their contralateral uninjured TAs (Fig. 2b). This indicated

that only the wild-type mice had recovered the totality of their muscle mass by 14 DPI. At this late time point, the _dy^W_/_dy^W_ muscles still contained high numbers of mono-nucleated cells and prominent interstitial space between regenerating fibers (Fig. 2d, orange arrows).

Over the injury time course, both mice saw an increase in the size of regenerated fibers (Figs. 2d, S3h–k). However, as it is already the case in uninjured TAs, the regenerated fibers of _dy^W_/_dy^W_ mice had a smaller diameter than the regenerated fibers of wild-type mice (Fig. S3h–k). Furthermore, while the difference was not statistically significant, wild-type mice showed a trend towards regenerating more fibers than _dy^W_/_dy^W_ mice. At 7 DPI, for example, wild-type mice contained on average 25% more fibers in their injured TA compared to their uninjured TA (Fig. 2c). In _dy^W_/_dy^W_ mice, the injured TA had approximately the same number of fibers as the untreated contralateral TA (Fig. 2c). Upon quantifying the number of centrally-located nuclei in regenerated fibers, it also became apparent that at 7 DPI, the _dy^W_/_dy^W_ TAs had a significantly lower proportion of fibers with at least 2 centrally-located nuclei (Fig. 2e, f). This indicated that less MuSC fusion had occurred by 7 DPI, as reductions in MuSC fusion have been shown to decrease the number of centralized nuclei in regenerated fibers[33]. By 14 DPI, this disparity was no longer detected (Fig. S3l), suggesting that MuSC fusion was delayed in _dy^W_/_dy^W_ mice.

To determine whether an abnormal MuSC response would contribute to the regenerative defects in _dy^W_/_dy^W_ mice, we stained and quantified Pax7+ and Myogenin+ cells in the TAs pre- and post-injury (Fig. 2g). While they were more abundant in _dy^W_/_dy^W_ muscle pre-injury (Figs. 2h, i, S3e), Pax7+ and Myogenin+ cells only reached their peak number at 7 DPI in _dy^W_/_dy^W_ mice (Fig. 2h, i), whereas in wild-type mice they had already peaked by 4 DPI (Fig. 2g–i). This delayed expansion of the _dy^W_/_dy^W_ MuSCs and their progeny was not due to higher levels of quiescence, as there were no significant differences in the proportion of Ki67+/Pax7+ cells at these time points (Fig. S3m).

All in all, characterization of wild-type and _dy^W_/_dy^W_ TAs post-injury highlighted that the LAMA2 MD mouse model exhibits regeneration deficits that include a delay in MuSC expansion. These findings are in line with other studies[8,9] and raise the possibility that _Lama2_ mutations directly impair MuSC function in LAMA2 MD.

## _dy^W_/_dy^W_ MuSCs proliferate more slowly than wild-type MuSCs ex vivo

To probe the origin of these regenerative defects and gain more insight into the MuSCs' properties, we used FACS to isolate MuSCs from wild-type and _dy^W_/_dy^W_ mice at 4 weeks of age. Isolated PMs were cultured ex vivo on collagen-coated tissue culture dishes to assess cell activation, proliferation and differentiation.

MuSC activation kinetics were measured by culturing the cells with EdU in the medium for 30 h post-sorting. Although there was no significant difference in the proportion of activated Ki67+/Pax7+ cells at this time point (Fig. 3a), the detection of EdU uncovered a major disparity in the PMs' cell-cycle kinetics. While approximately 91% of wild-type PMs had replicated their DNA during the first 30 h in culture,

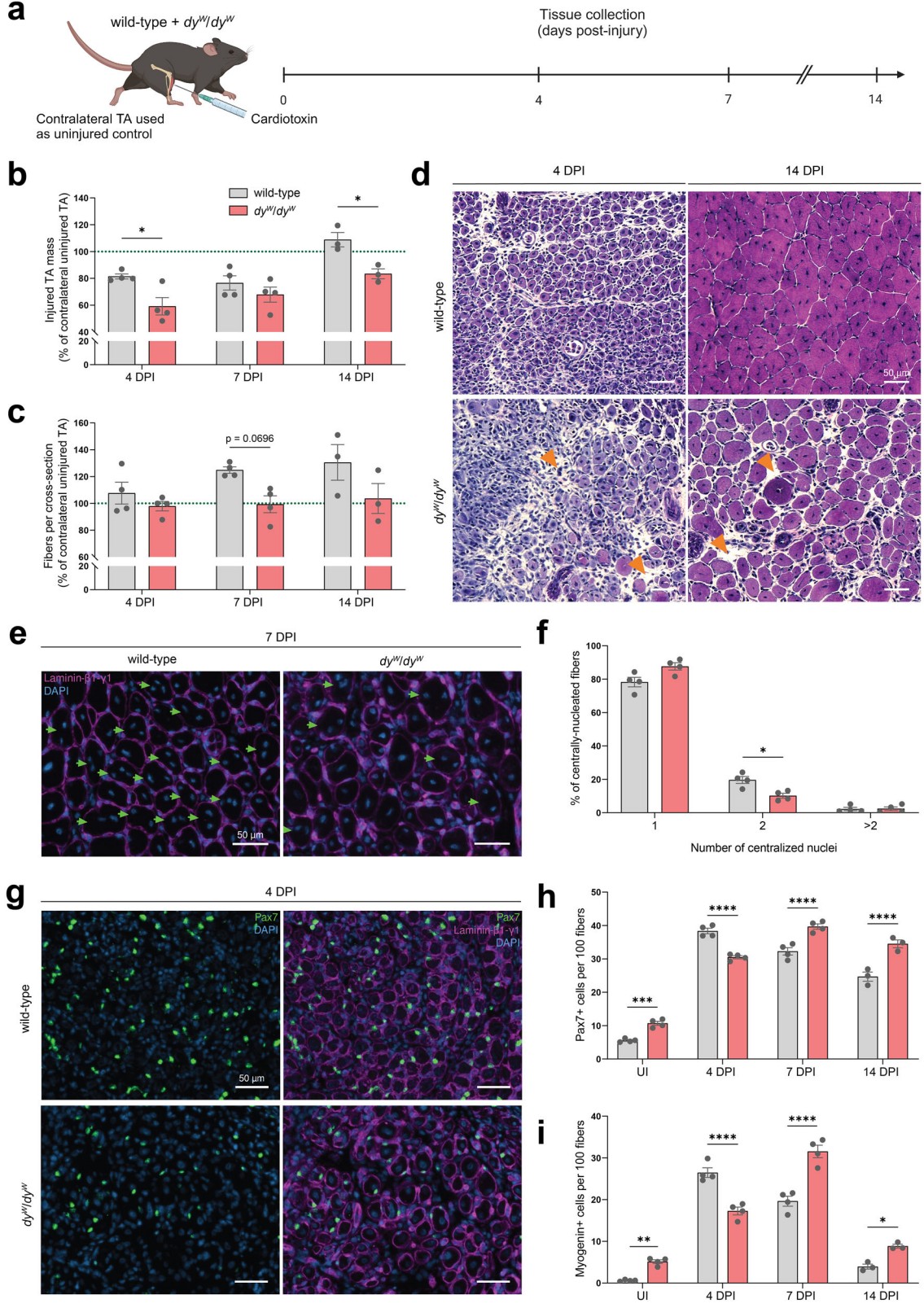

only 56% of the $dy^W/dy^W$ PMs had done so (Fig. 3a). This indicates that once they are activated, $dy^W/dy^W$ PMs progress through the cell-cycle at a slower rate. To test whether this phenotype persisted and was independent of cell density, we seeded cells at 100 and 200 cells/mm² after the fourth passage and added EdU to the medium for 15 h. Again, $dy^W/dy^W$ PMs incorporated less EdU than wild-type PMs (Fig. 3b, c). Flow cytometry-mediated cell-cycle analysis[34] confirmed that a smaller

proportion of $dy^W/dy^W$ PMs were in S phase due to their accumulation in G1 (Fig. 3d). Daily live imaging of cell density for 5 days after plating revealed that this slower cell-cycle progression in $dy^W/dy^W$ PMs led to a reduction in proliferation compared to wild-type cells (Fig. 3e). This was not due to poor cell attachment, as 1-day post-plating, there were no differences in the number of wild-type and $dy^W/dy^W$ PMs attached to the plate (Fig. 3e). It was also not due to higher levels of apoptosis, as

**Fig. 2 | Impaired muscle regeneration in mouse models for LAMA2 MD.**
**a** Experimental approach: TA muscles of wild-type and $dy^W/dy^W$ mice were injured by cardiotoxin injection. Contralateral TAs were used as uninjured (UI) controls. Both TAs were collected at 4, 7 and 14 DPI for analyses. **b** Quantification of TA masses at 4, 7 and 14 DPI. Injured TA masses are normalized to the contralateral uninjured TA's mass (green dotted line). $n = 4$ mice at 4 and 7 DPI; $n = 3$ mice at 14 DPI. **c** Quantification of the number of fibers per cross-section post-injury. Values are normalized to the number of fibers in the contralateral uninjured TA (green dotted line). $n = 4$ mice at 4 and 7 DPI; $n = 3$ mice at 14 DPI. **d** H&E staining of wild-type and $dy^W/dy^W$ TA cross-sections at 4 and 14 DPI. Orange arrows: regions with large interstitial space. **e** Representative immunostaining of centrally-located nuclei

at 7 DPI (laminin-β1-γ1 in magenta, DAPI in blue). Green arrows: fibers with 2 or more centrally-located nuclei. **f** Quantification of the number of centralized nuclei per centrally-nucleated fiber at 7 DPI (average of 3 sections per mouse; $n = 4$ mice). **g** Representative immunostaining of Pax7+ cells at 4 DPI (Pax7 in green, laminin-β1-γ1 in magenta, DAPI in blue). **h** and **i** Quantification of the number of Pax7+ (**h**) and Myogenin+ (**i**) cells per 100 fibers (average of 3 sections per mouse; $n = 4$ mice at 4 and 7 DPI; $n = 3$ mice at 14 DPI). Data are means ± SEM. Statistical significance was determined by two-way ANOVAs with Bonferroni's multiple comparisons test (**b**, **c**, **h**, and **i**) or unpaired Student's two-sided $t$-test (**f**). *$P < 0.05$; **$P < 0.01$; ***$P < 0.001$; ****$P < 0.0001$. Source data and $P$-values are provided as a Source Data file. **a** Created in BioRender. Ruegg, M. (2025) https://BioRender.com/d18khcw.

similar proportions of wild-type and $dy^W/dy^W$ PMs stained positive for cleaved caspase-3 after 5 days in culture (Fig. 3f).

To test the cells' myogenic differentiation capacity, we cultured them in differentiation medium for 7 days and stained them with desmin antibodies. At this time point, both wild-type and $dy^W/dy^W$ PMs had fused and formed multinucleated myotubes (Fig. 3g). Quantifications of the fusion index revealed no differences in the cells' differentiation capacities (Fig. 3h). To corroborate these findings, we isolated and cultured single fibers from the *gastrocnemius* (GAS) muscle of wild-type and $dy^W/dy^W$ mice. Fibers were fixed and stained with different markers at different time points: immediately post-isolation (T0), after 42 (T42) and 72 (T72) hours in culture (Fig. 3i). Consistent with our observations that MuSCs in $dy^W/dy^W$ mice are activated because of the constant cycles of muscle degeneration/regeneration (Fig. S3c), fibers from $dy^W/dy^W$ mice harbored a significantly higher proportion of activated Ki67+/Pax7+ MuSCs at T0 (Fig. 3j). By T42, this difference was erased as almost all wild-type and $dy^W/dy^W$ Pax7+ cells were Ki67+ (Fig. 3j). Though there were no differences in the proportion of activated Pax7+ cells at T42, a significantly higher proportion of wild-type Pax7+ cells had divided to form clusters containing 3- to 7 cells (Fig. 3i, k), whereas more than half of the $dy^W/dy^W$ Pax7+ MuSCs remained as single cells (Fig. 3k). This suggested that wild-type MuSCs had undergone more divisions than $dy^W/dy^W$ MuSCs during the first 42 h of culture. This phenotype was also evident at T72, as approximately 48% of wild-type Pax7+ cells were found in clusters comprising more than 7 cells, versus only 6% for $dy^W/dy^W$ cells (Fig. 3i, k). These results are consistent with our tissue culture observations that $dy^W/dy^W$ PMs proliferated more slowly than their wild-type counterparts (Fig. 3a–e).

Finally, we quantified the proportion of Pax7+ and Myogenin+ cells on the fibers at these different time points to assess MuSC differentiation capacities. Though the difference was not statistically significant, we observed a higher proportion of Myogenin+ cells on $dy^W/dy^W$ fibers at T0 (Fig. 3l). From T0 to T42, the proportion of Myogenin+ cells increased from 3% to 11% for wild-type mice and from 18% to 27% for $dy^W/dy^W$ mice (Fig. 3l). By T72, more than 25% of cells were Myogenin+ for both groups (Fig. 3i, l). While baseline differences in the number of Myogenin+ cells render comparisons difficult, these experiments confirm that wild-type and $dy^W/dy^W$ MuSCs can differentiate into Myogenin+ cells on fibers ex vivo.

All in all, ex vivo characterizations of wild-type and $dy^W/dy^W$ MuSCs revealed that $dy^W/dy^W$ cells suffer from a proliferation defect. Interestingly, it is during this proliferative state that wild-type MuSCs exhibit the strongest expression of *Lama2* transcripts and the highest production of laminin-α2 (Fig. 1a–i).

**Proliferative defects lower the contribution of $dy^W/dy^W$ MuSCs to tissue remodeling post-transplantation**

With mounting evidence that MuSC function is impaired in $dy^W/dy^W$ mice, we wanted to determine whether the origin of this phenotype was cell-intrinsic and/or cell-extrinsic. As it has recently been demonstrated that LAMA2 MD mouse models bear changes in muscle composition that can influence their regenerative capacities[8], we

hypothesized that transplanting $dy^W/dy^W$ MuSCs into a non-dystrophic muscle might rescue their function. If the disease-causing *Lama2* mutation intrinsically impairs $dy^W/dy^W$ MuSC function, then transplantation will not be sufficient to rescue their regenerative deficit.

To test this, we crossed wild-type and $dy^W/dy^W$ mice with mice that allow labeling of Pax7+ MuSCs with EGFP upon tamoxifen injection[29,30]. After 5 days of tamoxifen treatment, we isolated EGFP+ MuSCs from these 5-week-old mice via FACS and transplanted 10,000 cells into the CTX-injured TAs of immunodeficient NOD *scid* gamma (NSG) mice at 1 DPI. Recipient muscles were then collected at 3, 7 and 21 DPI to assess the contribution of EGFP+ MuSCs to tissue remodeling. At 3 DPI, mice were administered EdU 3 h prior to tissue collection to enable comparisons of cell-cycle kinetics (Fig. 4a).

Quantifications of the number of EGFP+ cells and fibers per cross-section revealed that at 3 DPI, there was no significant difference in the number of engrafted wild-type and $dy^W/dy^W$ MuSCs (Fig. 4b, c). By 7 DPI, mice that received wild-type EGFP+ MuSCs had significantly more EGFP+ cells and fibers per cross-section than mice that received $dy^W/dy^W$ EGFP+ MuSCs (Fig. 4b, c). The latter also contained fewer Pax7+ and Myogenin+ transplanted cells at this time point (Fig. 4d, e). By 21 DPI, the number of Pax7+ and Myogenin+ EGFP+ cells was no longer significantly different (Fig. 4d, e), but the discrepancy in the number of EGFP+ cells and fibers per cross-section became even more pronounced (Fig. 4b, c). Transplanted wild-type EGFP+ MuSCs yielded an average of 402 EGFP+ cells and fibers per cross-section, whereas $dy^W/dy^W$ EGFP+ MuSCs contributed substantially less by generating only 102 (Fig. 4c). Since this low contribution to tissue remodeling was not caused by a reduction in cell engraftment at 3 DPI (Fig. 4c), we characterized the cell-cycle state of transplanted MuSCs at the different time points. While no significant differences in the proportion of Ki67+ MuSCs were detected at 3 and 7 DPI (Fig. 4f, g), many wild-type MuSCs had regained quiescence by 21 DPI (17% Ki67+), whereas half of the $dy^W/dy^W$ MuSCs remained activated (47% Ki67+) (Fig. 4g). This persistence of activated cells cannot be explained by a failure to enter quiescence, as more than a third of the $dy^W/dy^W$ MuSCs exited the cell-cycle from 3 to 21 DPI (Fig. 4g). Slower cell-cycle progression, however, could result in lower contribution to tissue remodeling (Fig. 4b, c) and a deferral of the $dy^W/dy^W$ MuSCs' re-entry into quiescence (Fig. 4g). We therefore assessed the cell-cycle progression of transplanted wild-type and $dy^W/dy^W$ MuSCs at 3 DPI by counting the proportion of EdU+/EGFP+ cells. Quantifications revealed that significantly more wild-type cells had replicated their DNA in the 3 h prior to tissue collection (Fig. 4f, h), confirming that they were progressing through the cell-cycle faster than $dy^W/dy^W$ MuSCs. Since there were no differences in the proportion of apoptotic TUNEL+/EGFP+ cells (Fig. 4i), these results confirm that the lower tissue remodeling contribution of $dy^W/dy^W$ MuSCs stems from a slower cell-cycle progression, which impedes their expansion and delays their cell-cycle exit.

As it was shown that MuSCs exhibit a senescent phenotype in muscular dystrophies[35,36] and that transplantation of senescent cells blunts muscle regeneration[37], we wanted to determine if the transplantation of $dy^W/dy^W$ MuSCs had an overall negative impact on tissue

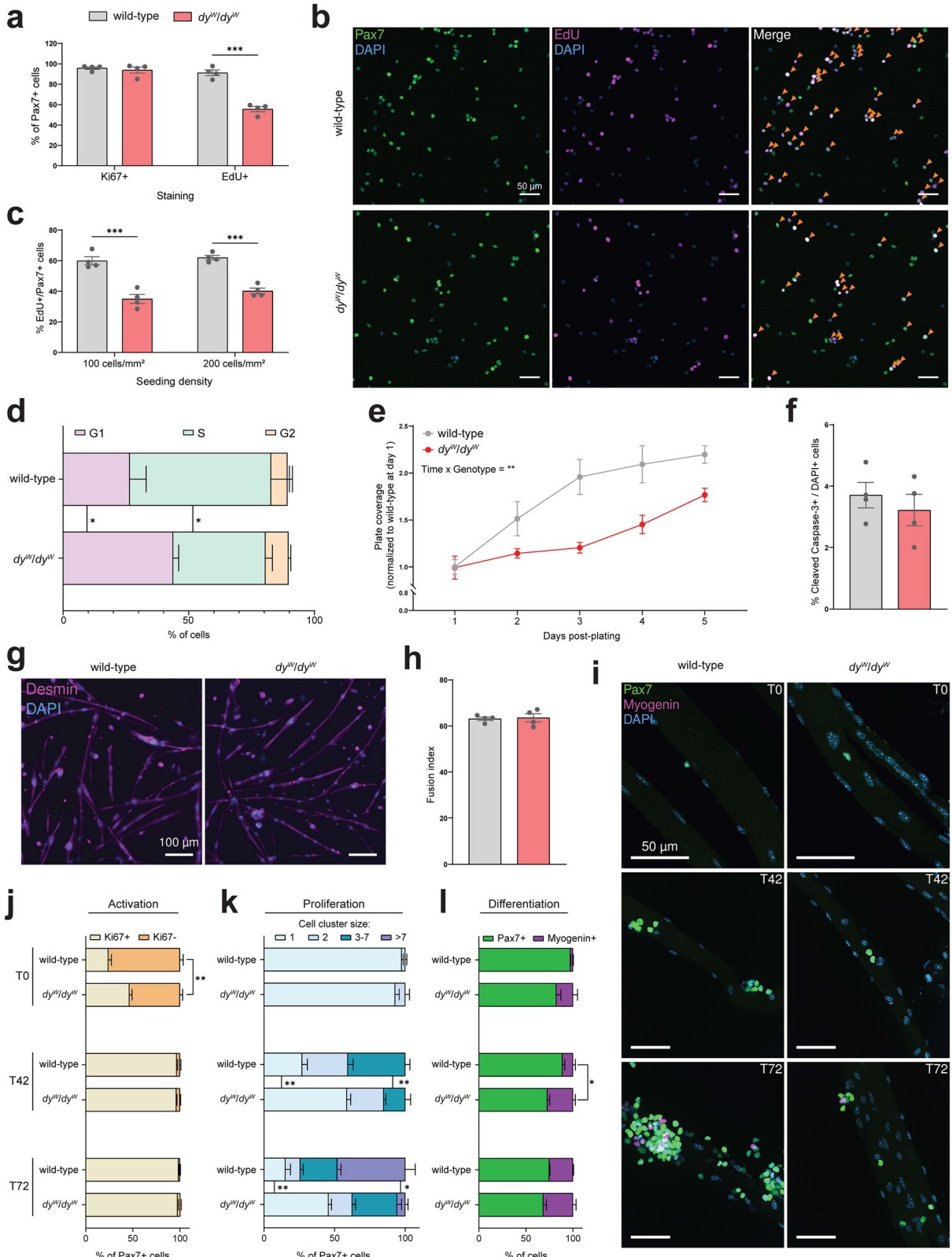

regeneration. Irrespective of whether NSG mice were injected with wild-type or $dy^W/dy^W$ EGFP+ MuSCs, the muscle mass recovery post-injury was the same (Fig. S4a). Histological analyses showed minimal differences between the transplanted TAs at 7 and 21 DPI (Fig. S4b), though there was a trend toward increased Sirius Red staining in mice that received $dy^W/dy^W$ MuSCs at 7 DPI (Fig. S4c). This slight increase in fibrosis did not correlate with a higher presence of Pdgfrα+FAPs or

F4/80+ macrophages (Fig. S4d). Finally, whether the mice received wild-type or $dy^W/dy^W$ MuSCs, regenerated EGFP+ fibers showed no differences in the fiber size distribution (Fig. S4e, S4f). The contribution of $dy^W/dy^W$ MuSCs to fiber regeneration, therefore, had no impact on fiber size. Similarly, the size of fibers regenerated solely from EGFP-negative, endogenous MuSCs did not differ from those that also included EGFP+ MuSCs, and this was irrespective of their origin (Fig.

**Fig. 3 | $dy^W/dy^W$ MuSCs proliferate more slowly than wild-type MuSCs ex vivo.** **a** Quantification of the proportion of Ki67+/Pax7+ and EdU+/Pax7+ cells 30 h post-sorting. **b** Representative immunostaining of PMs (after 4th passage) cultured with EdU for 15 h (Pax7 in green, EdU in magenta, DAPI in blue). Orange arrows: EdU+/Pax7+ cells. **c** Quantification of EdU+/Pax7+ cells seeded at the indicated density and labeled with EdU for 15 h. **d** Flow cytometry-mediated cell-cycle analysis[34] of wild-type and $dy^W/dy^W$ PMs in proliferation conditions. **e** Quantification of PM proliferation for 5 days after plating (after 4th passage). Each value is the average calculated from three 4x magnification images per well. **f** Quantification of the proportion of cleaved capase-3+ cells 5 days post-plating. **g** Representative immunostaining of myotubes after 7 days in differentiation medium (desmin in magenta, DAPI in blue). **h** Fusion index (number of nuclei in myotubes divided by the total number of nuclei) after 7 days in differentiation medium. **i** Representative immunostaining of single *gastrocnemius* fibers in culture for 0 (T0), 42 (T42) or 72 (T72) h (Pax7 in green, Myogenin in magenta, DAPI in blue). **j** Quantification of the proportion of Ki67+/Pax7+ cells at T0, T42 and T72. **k** Quantification of the size of Pax7+ cell clusters at T0, T42 and T72. Cell-cell contact between Pax7+ cells was a requisite to include them as a cluster of 2 or more cells. **l** Quantification of the proportion of Pax7+ and Myogenin+ cells at T0, T42 and T72. In **j**–**l**, a minimum of 15 fibers were analyzed per mouse at each time point; $n = 4$ mice. For cell culture experiments, cells were isolated from $n = 4$ mice (**a, c, f,** and **h**), $n = 5$ mice (**d**) or $n = 3$ mice (**e**). Data are means ± SEM. Statistical significance was determined by unpaired Student's two-sided *t*-test (**a, c, d, f,** and **h**) or two-way ANOVAs with Bonferroni's multiple comparisons test (**e, j, k,** and **l**). *$P < 0.05$; **$P < 0.01$; ***$P < 0.001$. Source data and *P*-values are provided as a Source Data file.

S4e, S4f). These results show that transplanted $dy^W/dy^W$ MuSCs do not affect overall regeneration.

In summary, these experiments revealed that $dy^W/dy^W$ MuSCs have a lower regenerative potential than wild-type MuSCs post-transplantation due to a slower cell-cycle progression and hence diminished expansion. This indicates that the $dy^W/dy^W$ MuSCs' proliferative defects are not restored by their removal from the dystrophic milieu. Furthermore, it suggests that their regenerative impairments cannot be attributed to the absence of laminin-α2 in the muscle BM, as their slower cell-cycle progression persisted in spite of their engraftment in a laminin-α2-rich environment. However, since laminin-α2 is secreted by wild-type MuSCs (Fig. 1i), we hypothesized that transplanted *Lama2*-deficient $dy^W/dy^W$ cells might be exposed to a laminin-α2-deficient microenvironment. Indeed, immunostaining revealed that at 3 DPI, during the proliferation of transplanted MuSCs in the interstitial space between BM remnants of ghost fibers, wild-type but not $dy^W/dy^W$ MuSCs co-localized with laminin-α2 (Fig. S5). By 7 DPI, laminin-α2 was occasionally in the vicinity of $dy^W/dy^W$ MuSCs but at much lower levels than around the wild-type MuSCs (Fig. S5). At 21 DPI, all EGFP+ fibers were similarly surrounded by laminin-α2 (Fig. S5). Taken together, these observations suggest that at the early time point, proliferating wild-type MuSCs deposit laminin-α2 in their microenvironment while $dy^W/dy^W$ MuSCs are incapable of doing so. We therefore hypothesized that a loss of self-secreted laminin-α2 could underlie the $dy^W/dy^W$ MuSCs' proliferative defects.

## MuSC-specific *Lama2* knockout is sufficient to slow proliferation and delay regeneration

While the results so far strongly indicate that *Lama2*-deficient MuSCs have a cell-intrinsic proliferation deficit, it remains possible that MuSCs isolated from $dy^W/dy^W$ mice are still hampered by their prior exposure to a dystrophic environment. To circumvent these extrinsic factors and to test whether loss of *Lama2* expression in MuSCs directly impairs proliferation, we generated mice that carry loxP sites in the introns flanking exon 3 of *Lama2*. Deletion of exon 3—which is commonly observed in LAMA2 MD patients[38]—results in frameshift mutations and the introduction of stop codons from exon 4 onwards[39]. We crossed these mice with Pax7[CreER] mice[29] and mice in which a CAG-promoter, floxed-stop cassette and EGFP sequence are knocked into the *Rosa 26* locus[30]. Hence, after tamoxifen treatment, MuSCs in these mice will be *Lama2*-deficient and positive for EGFP (Fig. 5a). In the following experiments, MuSC-specific, inducible *Lama2* knockout mice (MuSC-*Lama2*KO) were homozygous for the floxed *Lama2* allele, while control littermates were wild-type for the *Lama2* allele (control). Both mice were heterozygous for Pax7[CreER] and the EGFP knock-in (Fig. 5a).

To test the efficacy of the system, we treated 4-week-old mice for 5 consecutive days with tamoxifen. Three days after the last injection, we collected the muscles and quantified the proportion of Pax7+ cells that were EGFP+ in cross-sections of the TA and on single fibers isolated from the EDL. For control and MuSC-*Lama2*KO mice, the 5 days of tamoxifen treatment resulted in approximately 90% and 95% of Pax7+

cells being EGFP+ in the TA and EDL, respectively (Fig. S6a). We then used FACS to isolate EGFP+ MuSCs from both mice and confirmed by PCR using primers flanking *Lama2*'s exon 3 that it was deleted in the EGFP+ cells from MuSC-*Lama2*KO mice, but not control mice (Fig. 5b). Finally, we also tested for the presence of laminin-α2 in tissue culture dishes containing PMs from both mice by staining the cultured cells with an anti-laminin-α2 antibody. While MuSC-*Lama2*KO and negative control dishes—where we omitted the anti-laminin-α2 antibody—were negative for laminin-α2 immunoreactivity, there was a clear laminin-α2 signal in dishes containing control PMs (Fig. 5c). This confirmed that control PMs produce laminin-α2 while MuSC-*Lama2*KO PMs are incapable of doing so. To measure the timing of cell-cycle progression in the two PM populations, we plated the cells on collagen-coated tissue culture dishes and labeled them with EdU at different time points. When EdU was added for the initial 6 h after plating, there was no difference in the proportion of EdU+ cells between control and *Lama2*-deficient PMs (Fig. 5d). However, when labeling was initiated after 6 h in culture and cells were analyzed 15 h later, significantly fewer MuSC-*Lama2*KO PMs incorporated EdU compared to the controls (Figs. 5d, S6b), indicating that they were progressing through the cell-cycle at a slower rate. To test whether *Lama2* deficiency would affect differentiation/fusion, we cultured the cells in differentiation medium for 3 days and quantified the proportion of nuclei residing in multi-nucleated myotubes. This showed that control and MuSC-*Lama2*KO EGFP+ cells differentiated and fused with similar efficacy (Figs. S6c, S6d). These results are consistent with those obtained with $dy^W/dy^W$ PMs, which showed a slowing of proliferation but normal differentiation.

Next, we tested the proliferation and differentiation of MuSCs in their niche using single fibers isolated from the EDL muscles of control and MuSC-*Lama2*KO mice. Fibers were stained immediately post-isolation (T0), after 24 (T24), 42 (T42) or 72 (T72) hours in culture (Fig. 5e). When staining for the activation marker Ki67, we noted no significant differences in the cell-cycle entry of control and MuSC-*Lama2*KO EGFP+ cells, although controls tended to have a higher proportion of Ki67+ cells at T0 and T24 (Fig. 5e, f). However, we observed significant disparities in the cells' proliferative capacities (Fig. 5e, g). At T0 and T24, most control and MuSC-*Lama2*KO EGFP+ cells were present on the fibers as single cells (Fig. 5e, g). By T42, approximately 32% of control EGFP+ cells were found in clusters containing 3- to 7 cells, versus 19% for the MuSC-*Lama2*KO. At this time point, 45% of MuSC-*Lama2*KO EGFP+ cells remained as single cells, versus only 21% for the controls (Fig. 5g). This reduction in proliferation was also apparent at T72, when more than 33% of control EGFP+ cells were found in clusters comprising more than 7 cells, while for the MuSC-*Lama2*KO this number was 14% (Fig. 5g). Interestingly, though quiescent control and MuSC-*Lama2*KO EGFP+ cells were located below laminin-α2 at T0, from T24 onwards, we observed abundant laminin-α2 signal around control cells, but none around *Lama2*-deficient MuSCs (Fig. 5e). This confirms the effectiveness of the *Lama2* knockout and reinforces that the laminin-α2 surrounding proliferating

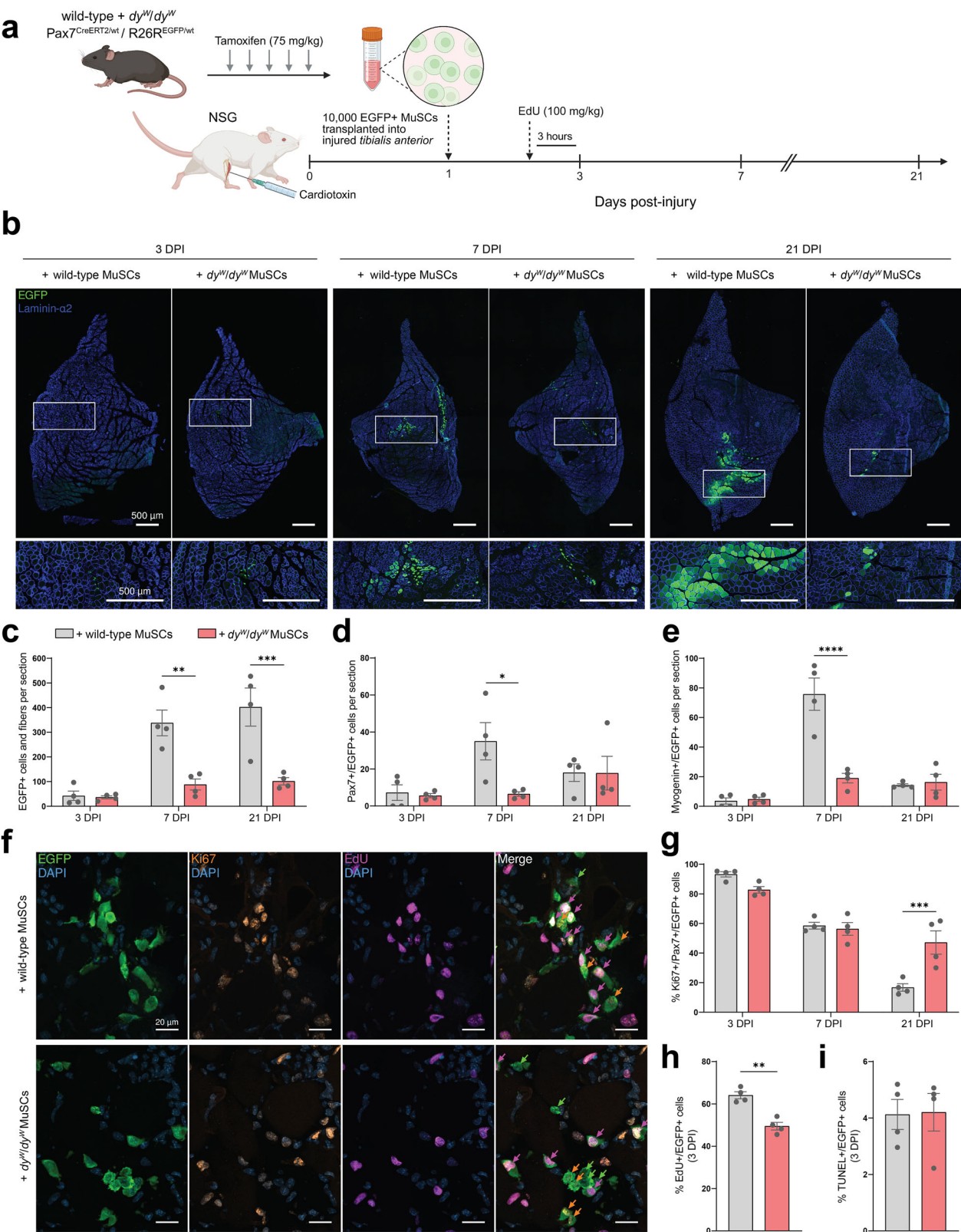

MuSCs is self-secreted. Finally, to determine if the MuSC-*Lama2*KO influences MuSC differentiation, we stained the fibers with Pax7 and Myogenin at the different time points. Overall, we observed that from T0 to T72, the proportion of Pax7+/EGFP+ cells decreased and the proportion of Myogenin+/EGFP+ cells increased for the controls and MuSC-*Lama2*KOs (Fig. 5h). Though there was a trend towards higher proportions of Myogenin+/EGFP+ cells in the controls at T72, the

difference was not statistically significant (Fig. 5h). There were also no significant differences in the proportion of MyoD+/EGFP+ cells at T72 (Fig. S6e), indicating that MuSC self-renewal and differentiation were not influenced by the loss of laminin-α2 secretion. Taken together, these experiments demonstrate that *Lama2*-deficient MuSCs show a proliferation deficit—but no differentiation defects—when they are still attached to their fibers.

**Fig. 4 | Proliferative defects hinder the contribution of $dy^W/dy^W$ MuSCs to tissue remodeling post-transplantation. a** Experimental approach: EGFP+ MuSCs were isolated from wild-type and $dy^W/dy^W$ mice via FACS, and 10,000 cells were transplanted into the injured TAs of NSG mice at 1 DPI. Recipient muscles were collected at 3, 7 and 21 DPI for analyses. Mice were administered EdU 3 h before tissue harvesting at 3 DPI. **b** Representative immunostaining of recipient TAs at 3, 7 and 21 DPI (EGFP in green, laminin-α2 in blue). **c** Quantification of the number of EGFP+ cells and fibers per cross-section (average of 20 sections per mouse at 3 DPI; average of 3 sections per mouse at 7 and 21 DPI; $n = 4$ mice). **d** and **e** Quantification of the number of Pax7+/EGFP+ (**d**) and Myogenin+/EGFP+ (**e**) cells per cross-section (average of 3 sections per mouse; $n = 4$ mice). **f** Representative immunostaining of recipient TAs at 3 DPI (EGFP in green, Ki67 in orange, EdU in magenta, DAPI in blue).

EGFP+/Ki67−/EdU− (green arrows), EGFP + /Ki67 + /EdU- (orange arrows) and EGFP +/Ki67+/EdU+ (magenta arrows) cells are indicated in the merge panel. **g** Quantification of the proportion of Ki67+/Pax7+/EGFP+ cells (average of 10 sections per mouse at 3 DPI; average of 3 sections per mouse at 7 and 21 DPI; $n = 4$ mice). **h** Quantification of the proportion of EdU+/EGFP+ cells at 3 DPI (average of 10 sections per mouse; $n = 4$ mice). **i** Quantification of the proportion of apoptotic TUNEL+/EGFP+ cells at 3 DPI (average of 10 sections per mouse; $n = 4$ mice). Data are means ± SEM. Statistical significance was determined by two-way ANOVAs with Bonferroni's multiple comparisons test (**c**–**e** and **g**) or by unpaired Student's two-sided $t$-test (**h** and **i**). *$P < 0.05$; **$P < 0.01$; ***$P < 0.001$; ****$P < 0.0001$. Source data and $P$-values are provided as a Source Data file. **a** Created in BioRender. Ruegg, M. (2025) https://BioRender.com/d18khcw.

Next, we tested whether *Lama2*-deficient MuSCs exhibit deficits in muscle regeneration upon acute injury. Three days after the last tamoxifen injection, the TA muscle of control and MuSC-*Lama2*KO mice was injured by injection of CTX (Fig. 6a). Injured TAs and the contralateral, uninjured TAs were analyzed at 4 and 10 DPI. As baseline controls, we also collected TA muscles from tamoxifen-treated mice that did not receive CTX. Upon tissue collection, we observed that the injured TA's relative mass was significantly lower in MuSC-*Lama2*KO mice at 4 DPI, but not at 10 DPI (Fig. 6b). While there were no differences in the proportion of Ki67+ MuSCs post-injury (Fig. 6c, d), we observed that Pax7+ and Myogenin+ cell expansion was significantly hindered in MuSC-*Lama2*KO mice at 4 DPI (Fig. 6c, e, f). This confirmed that MuSC-specific *Lama2* knockout delays MuSC expansion post-injury.

While this delayed expansion had no influence on the levels of fibrosis (Fig. S6f), the number of centrally-located nuclei (Fig. S6g), or the size (Fig. S6h–j) and the relative number of fibers post-injury (Fig. 6g), we observed that TAs from control mice contained a higher proportion of fully-fused EGFP+ fibers than TAs from MuSC-*Lama2*KO mice at 4 DPI (Fig. 6h, i), though the difference did not reach statistical significance. Interestingly, we detected laminin-α2 on the basal and apical sides of EGFP+ cells within the partially-fused EGFP+ fibers of control mice, but not those of MuSC-*Lama2*KO mice (Figs. 6h, S6k, yellow arrows). We also observed that interstitially-located EGFP+ cells from control mice co-localized with laminin-α2, while those from MuSC-*Lama2*KO mice did not (Fig. S6k, orange arrows). This suggested that, despite the presence of *Lama2*-expressing muscle-resident cells, muscles from MuSC-*Lama2*KO mice may contain lower amounts of laminin-α2 post-injury. To measure this, we quantified the relative abundance of laminin-α2 in the injured TAs of control and MuSC-*Lama2*KO mice, and normalized it to the amount detected in the contralateral, uninjured TA of the same mouse. As reported above (Fig. S2b), laminin-α2 levels in control mice tended to be increased in injured TAs at 4 DPI (Fig. S6l, S6m). At this time point, MuSC-*Lama2*KO mice did not show this increase, resulting in significantly lower relative laminin-α2 levels compared to control mice (Fig. S6l, S6m). We confirmed this in muscle cross-sections, as laminin-α2 signal intensity was significantly reduced in MuSC-*Lama2*KO mice compared to control mice at 4 DPI (Fig. S6n, S6o). By 10 DPI, there was no longer a difference in the relative abundance of laminin-α2 shown by Western blot (Fig. S6l, S6m) or by immunofluorescence (Fig. S6n, S6o). Since there were also no differences in laminin-α2 abundance in uninjured tissue (Fig. S6p), these results indicate that MuSCs are sources of laminin-α2 during the early stages of regeneration. The recovery of relative laminin-α2 abundance observed at 10 DPI in MuSC-*Lama2*KO mice (Fig. S6m, S6o) suggests that other cells eventually compensate for the loss of MuSC-secreted laminin-α2.

To confirm that MuSC-specific *Lama2* knockout affects proliferation with a second knockout approach, we adapted a CRISPR-based method that was previously developed to knockout genes in muscle fibers[40] for the knockout of *Lama2* in isogenic MuSCs ex vivo. For this, we used mice in which tamoxifen administration leads to

MuSC-specific expression of Cas9 and EGFP, separated by a self-cleaving P2A peptide[29,41] (Fig. S7a). We then used FACS to isolate EGFP+ MuSCs from these 5-week-old mice and expanded the cells on collagen-coated tissue culture dishes. For each mouse, EGFP+ Cas9-expressing cells were treated with AAVMYO[42], an adeno-associated virus that targets muscle fibers and MuSCs[43], containing either a CMV-driven tdTomato transgene or a CMV-driven tdTomato transgene and the sequence for 3 single guide RNAs (sgRNAs) targeting exons 2 and 3 of *Lama2*. After sorting transduced EGFP+/tdTomato+ cells (Fig. S7a), we observed *Lama2* DNA editing only in cells that received the sgRNAs (Fig. S7b). As expected, these cells did not express laminin-α2 (Fig. S7c). Most importantly, the knockout cells also showed slower cell-cycle progression compared to their isogenic controls (Fig. S7d, S7e) but no difference in myogenic differentiation (Fig. S7f). These results thus reaffirm that MuSC-specific *Lama2* knockout slows MuSC proliferation without affecting differentiation.

Since it was shown that Erk1/2 and Akt are downstream signaling effectors of ECM-binding receptors in MuSCs[44], we measured their phosphorylation levels in wild-type, $dy^W/dy^W$, control and MuSC-*Lama2*KO PMs (Fig. S8a). These quantifications revealed a strong downregulation of Erk1/2 phosphorylation in $dy^W/dy^W$ PMs, and a slight downregulation of both Erk1/2 and Akt phosphorylation in MuSC-*Lama2*KO PMs (Fig. S8a, S8b). As similar phenotypes were observed in β1-integrin-deficient MuSCs[44], we hypothesized that self-secreted laminin-α2 might support MuSC proliferation by binding β1-integrin-containing receptors and activating downstream mitogenic signaling pathways. Therefore, we measured β1-integrin activation by staining PMs with the monoclonal antibody 9EG7 that detects β1-integrin in its active conformation[45]. While we observed a strong trend towards lower active β1-integrin in $dy^W/dy^W$ PMs ($P = 0.0512$) (Fig. S8c, S8d), the abundance of active β1-integrin was unchanged in MuSC-*Lama2*KO PMs (Fig. S8c, S8d). Since *Itgb1* mRNA levels were also unchanged in *Lama2*-deficient PMs (Fig. S8e), these data indicate that while there is a lower activation of mitogenic signaling pathways in *Lama2*-deficient MuSCs, the proliferative influences of self-secreted laminin-α2 are not exclusively dependent on the activation of β1-integrin-containing receptors.

Finally, as the loss of laminin-α2 leads to an upregulation of laminin-α4 in the muscles of LAMA2 MD patients and LAMA2 MD mouse models[4], we examined whether *Lama2*-deficient MuSCs would also upregulate laminin-α4 secretion. While we detected abundant laminin-α4 surrounding the muscle fibers of $dy^W/dy^W$ mice at 14 DPI (Fig. S9a, green arrows), the laminin-α4 present in the muscles of wild-type, control and MuSC-*Lama2*KO mice was mostly found in puncta, indicative of blood vessels (Fig. S9a, magenta arrows). Hence, laminin-α4 abundance was increased in the muscles of $dy^W/dy^W$ mice but not in MuSC-*Lama2*KO mice (Fig. S9b). To determine if this increase in laminin-α4 was due to an upregulation of *Lama4* expression in MuSCs, we performed smRNA FISH on injured TAs of wild-type and $dy^W/dy^W$ mice at 4 DPI. Although *Lama4* mRNA abundance was increased in $dy^W/dy^W$ muscles (Fig. S9c), the transcript rarely co-localized with *Pax7*-expressing cells (Fig. S9c, white arrows),

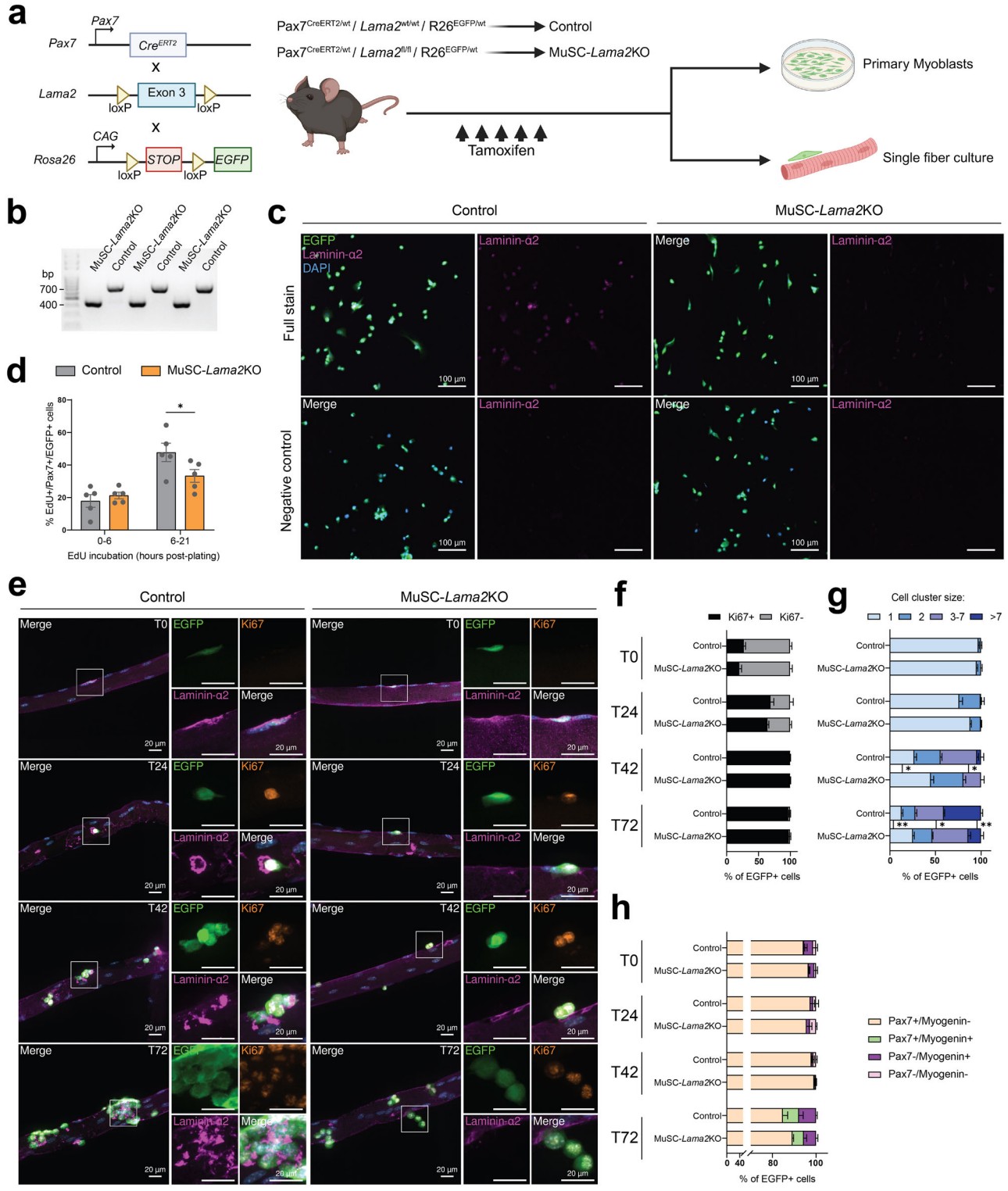

suggesting that the laminin-α4 detected in these mice (Fig. S9a) has a different cellular origin. To confirm that *Lama2*-deficient MuSCs were not upregulating laminin-α4 secretion, we stained the protein in cell culture dishes containing wild-type, $dy^W/dy^W$, control and MuSC-*Lama2*KO PMs. This revealed very little to no signal for all samples (Fig. S9d), thereby confirming that laminin-α4 production was not increased in *Lama2*-deficient PMs.

In summary, MuSCs that are depleted for laminin-α2 exhibit impaired proliferation, delayed muscle regeneration and disrupted ECM remodeling post-injury. These data indicate that the disease-

causing mutations intrinsically compromise MuSC function in LAMA2 MD, which may contribute to the disease phenotype.

## *LAMA2* knockout impairs the proliferation of human myogenic precursor cells

To determine if laminin-α2-mediated remodeling of the microenvironment is a conserved property of MuSCs, we analyzed an snRNA-seq dataset generated from healthy human muscle[46]. In this dataset, we found that *LAMA2* is expressed by human MuSCs (Fig. 7a). We also confirmed the expression of *LAMA2* by human MuSCs with a

**Fig. 5 | MuSC-specific *Lama2* knockout slows MuSC proliferation ex vivo.**
**a** Experimental approach: mice with MuSC-specific *Lama2* knockout (MuSC-*Lama2*KO) were homozygous for the floxed *Lama2* allele, while control littermates were wild-type for the *Lama2* allele (control). Both were heterozygous for Pax7[CreER] and the EGFP knock-in. Mice were treated for 5 consecutive days with tamoxifen and collected 3 days later for isolation of EGFP+ cells and single fibers. **b** PCR on DNA isolated from sorted EGFP+ control and MuSC-*Lama2*KO cells. Product size after deletion of exon 3 = 433 bp; product size of wild-type allele = 755 bp. **c** Representative anti-laminin-α2 staining in tissue culture dishes with (upper panels, full stain) or without (lower panels, negative control) secondary antibodies (EGFP in green, laminin-α2 in magenta, DAPI in blue). **d** Quantification of the proportion of EdU+/Pax7+/EGFP+ after incubation with EdU in the medium from 0–6 and 6–21 h in culture. Cells were isolated from *n* = 5 mice. **e** Representative immunostaining of single EDL fibers isolated from control and MuSC-*Lama2*KO mice and stained after 0 (T0), 24 (T24), 42 (T42) and 72 (T72) h in culture (EGFP in green, Ki67 in orange, laminin-α2 in magenta, DAPI in blue). **f** Quantification of the proportion of Ki67+ and Ki67− EGFP+ cells at T0, T24, T42 and T72. **g** Quantification of the size of clusters formed by dividing EGFP+ cells at T0, T24, T42 and T72. Cell-cell contact between EGFP+ cells was a requisite to include them as a cluster of 2 or more cells. **h** Quantification of the proportion of Pax7+/Myogenin−, Pax7+/Myogenin+, Pax7−/Myogenin+ and Pax7−/Myogenin− EGFP+ cells at T0, T24, T42 and T72. In **f–h**, a minimum of 20 fibers were analyzed per mouse at each time point; *n* = 4 mice. Data are means ± SEM. In all graphs, statistical significance was determined by two-way ANOVAs with Bonferroni's multiple comparisons test. *P < 0.05; **P < 0.01. Source data and *P*-values are provided as a Source Data file. **a** Created in BioRender. Ruegg, M. (2025) https://BioRender.com/d18khcw.

second source: the Genotype-Tissue Expression (GTEx) Portal (dbGaP accession number phs000424.v9.p2). This implies that disease-causing *LAMA2* mutations could provoke cell-intrinsic MuSC impairments in LAMA2 MD patients.

Given the severity of LAMA2 MD, we opted to study the impact of *LAMA2* mutations on human MuSCs via a modeling approach that does not rely on patient biopsies and that enables comparisons of isogenic control and *LAMA2* knockout cells. For this, we used human induced pluripotent stem cells (hiPSCs) from a healthy donor in which exon 3 of the *LAMA2* gene was deleted, which is also the strategy we used to generate the murine MuSC-*Lama2*KO (Fig. 5a, b), and also results in the loss of *LAMA2* mRNA (Fig. S10) and laminin-α2 protein[47]. We refer to this line as *LAMA2* Ex3 KO. We also generated a second *LAMA2* knockout hiPSC line (referred to as *LAMA2* Ex7 KO) by introducing a frameshift mutation into exon 7, which leads to premature stop codons and the loss of *LAMA2* mRNA (Fig. S10). Control, *LAMA2* Ex3 KO and *LAMA2* Ex7 KO hiPSCs were subsequently differentiated via a well-established myogenic differentiation protocol[48–50] to generate hiPSC-derived myogenic precursor cells (Fig. 7b).

To test whether deletion of *LAMA2* influenced the cell-cycle progression of human myogenic precursors, we performed a 24 h EdU chase with control, *LAMA2* Ex3 KO and *LAMA2* Ex7 KO myogenic precursors derived from 3 independent differentiations. As we had seen with $dy^W/dy^W$ (Fig. 3b, c) and MuSC-*Lama2*KO PMs (Figs. 5d, S6b), *LAMA2*-deficient myogenic precursors showed a reduction in EdU incorporation compared to the isogenic control (Fig. 7c, d). This indicated that deletion of *LAMA2* in human myogenic precursor cells is sufficient to slow cell-cycle progression. When the myogenic differentiation protocol was continued, all cell lines differentiated and fused to form multinucleated myotubes (Fig. 7e). This is also in line with our previous observations that *Lama2*-deficient murine PMs differentiate normally ex vivo (Figs. 3g, h, S6c, S6d).

To gain more insight into the influence of *LAMA2* knockouts at the transcriptional level, we performed next-generation RNA-sequencing of control and *LAMA2* Ex3 KO myogenic precursors derived from four independent differentiations. Principal Component (PC) analysis revealed that control and *LAMA2* Ex3 KO cells clustered separately along PC1 (Fig. 7f), while samples that were differentiated in parallel clustered along PC2 (Fig. 7f). Differential expression analysis showed significant changes in *LAMA2* Ex3 KO cells compared to the isogenic control (Fig. 7g), including the upregulation of *POSTN*, a gene that is also upregulated in mouse models of LAMA2 MD[51]. *LAMA2* Ex3 KO cells also showed a strong downregulation of *RPS4Y1* and *CHCHD2*, which encode for ribosomal protein S4 and a mitochondrial protein, respectively. While these genes have not been described in the context of muscular dystrophies, their downregulation suggests that *LAMA2* Ex3 KO may influence ribosomal function and cell metabolism in myogenic precursors. To better perceive the influences of these transcriptional changes, we performed Gene Set Enrichment Analysis[52]. This approach highlighted that genes related to the "p53 pathway" were upregulated in *LAMA2* Ex3 KO myogenic precursors

(Fig. 7h), while genes regulating "G2/M checkpoint", "E2F cell-cycle related targets" and "mitotic spindle assembly", were strongly downregulated (Fig. 7h). Taken together, the transcriptomic characterizations of control and *LAMA2* Ex3 KO human myogenic precursors revealed that the loss of *LAMA2* expression induces transcriptional changes that are consistent with impairments in cell-cycle progression. Whether these transcriptional changes are causal or consequential remains to be determined.

In summary, our results demonstrate that cell-autonomous remodeling of the ECM with laminin-α2 is a conserved property of murine and human MuSCs. In both species, loss-of-function mutations in the laminin-α2-encoding gene are sufficient to slow the cell-cycle progression of myogenic precursor cells. Collectively, these findings implicate MuSCs in the pathophysiology of LAMA2 MD by demonstrating that the disease-causing mutations intrinsically impair MuSC function.

## Discussion

In this manuscript, we provide compelling evidence that MuSC-derived laminin-α2 is essential for MuSC proliferation and thereby contributes to muscle regeneration. First, we show that *Lama2*-deficient MuSCs of $dy^W/dy^W$ mice exhibit proliferation defects ex vivo (Fig. 3a–e, i, k) and in vivo (Fig. 2g–i). Although previous studies have linked regenerative deficits in these mice to cell-extrinsic factors[8], we reveal that restricting *Lama2* deletion to MuSCs is sufficient to impair their proliferation (Figs. 5d, e, g, 6c, e, f, S6b). Therefore, our data indicate that MuSC function is directly compromised in LAMA2 MD.

Our results are hence similar to those obtained in DMD mouse models, where deficits in muscle regeneration were solely attributed to the constitutive activation of MuSCs in response to muscle fiber degeneration[53,54]. As a result, MuSC dysfunction was only considered to be a secondary consequence of muscle fiber frailty. Eventually, this view changed as studies revealed that DMD-causing mutations impaired MuSC function in an intrinsic manner[24]. This added to the understanding of DMD pathology by demonstrating that muscle wasting is the additive outcome of muscle fiber frailty and MuSC dysfunction, suggesting that improving MuSC function is an additional therapeutic target[25]. Other muscular dystrophies that may be subjected to similar impairments include Facioscapulohumeral dystrophy Type 1 (FSHD1) and muscular dystrophy, congenital, Lmna-related (MDCL), which share the commonality of being caused by mutations in genes that are expressed by MuSCs[55].

### The role of laminin-α2 in MuSC proliferation

Following CTX injury in wild-type mice, the majority of MuSCs proliferate along BM remnants of ghost fibers[31,32]. In this article, we show that laminin-α2 is present in the BM remnants of regenerating muscle, but it also surrounds MuSCs located within ghost fibers (Fig. 1g, h, Supplementary Movie 1) and those migrating through the interstitial space (Fig. 1h, Supplementary Movie 1). Since it embeds proliferating MuSCs, we initially hypothesized that laminin-α2, irrespective of its

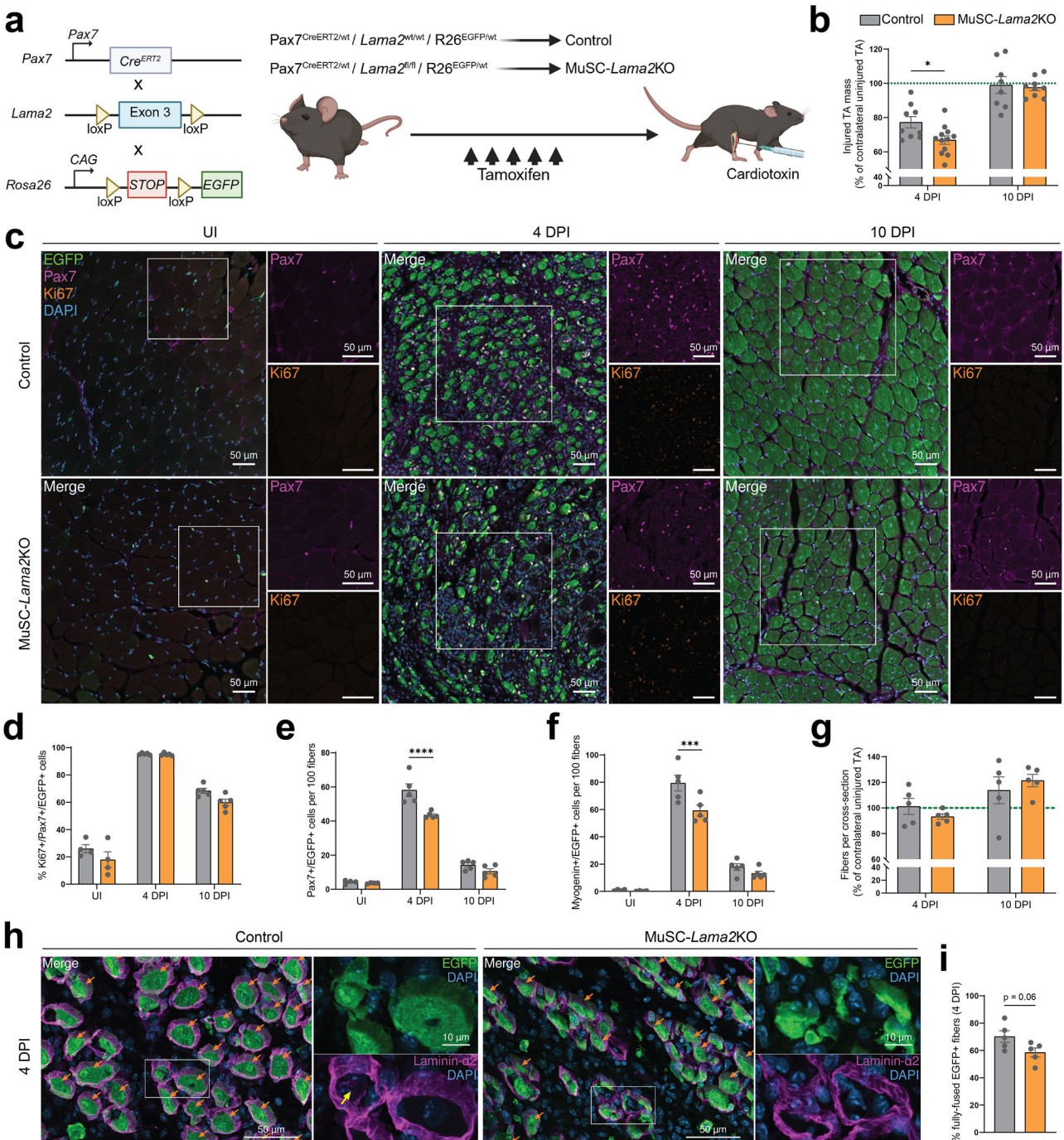

**Fig. 6 | MuSC-specific *Lama2* knockout delays regeneration. a** Experimental approach: three days after the last tamoxifen injection, TA muscles from control and MuSC-*Lama2*KO mice were collected as uninjured (UI) controls, or they were injured by cardiotoxin injection and collected at 4 and 10 DPI. **b** Quantification of TA masses at 4 and 10 DPI. Injured TA masses are normalized to the contralateral uninjured TA's mass (green dotted line). At 4 DPI, $n = 9$ control mice; $n = 12$ MuSC-*Lama2*KO mice. At 10 DPI, $n = 8$ control mice; $n = 9$ MuSC-*Lama2*KO mice. **c** Representative immunostaining of control and MuSC-*Lama2*KO TA cross-sections in uninjured conditions, and at 4 and 10 DPI (EGFP in green, Pax7 in magenta, Ki67 in orange, DAPI in blue). **d** Quantification of the proportion of Ki67+/Pax7+/EGFP+ cells pre- and post-injury (average of 3 sections per mouse). **e** and **f** Quantification of the number of Pax7+/EGFP+ (**e**) and Myogenin+/EGFP+ (**f**) cells per 100 fibers pre- and post-injury (average of 3 sections per mouse). **g** Quantification of the relative

number of fibers per cross-section post-injury, shown as a percentage of the number of fibers in the contralateral uninjured TA (green dotted line). **h** Representative immunostaining of control and MuSC-*Lama2*KO TA cross-sections at 4 DPI (EGFP in green, laminin-α2 in magenta, DAPI in blue). Orange arrows: fully-fused muscle fibers that are homogenously filled with EGFP and surrounded by a basement membrane; yellow arrow: presence of laminin-α2 on the apical side of EGFP+ cells in ghost fibers. **i** Quantification of the proportion of fully-fused EGFP+ fibers at 4 DPI (average of 3 sections per mouse). In **d**–**f**, $n = 4$ mice in uninjured conditions. In **d**–**g**, and **i**, $n = 5$ mice at 4 and 10 DPI. Data are means ± SEM. Statistical significance was determined by unpaired Student's two-sided *t*-test (**i**) or two-way ANOVAs with Bonferroni's multiple comparisons test (**b**, **d**–**g**). *$P < 0.05$; ***$P < 0.001$; ****$P < 0.0001$. Source data and *P*-values are provided as a Source Data file. **a** Created in BioRender. Ruegg, M. (2025) https://BioRender.com/d18khcw.

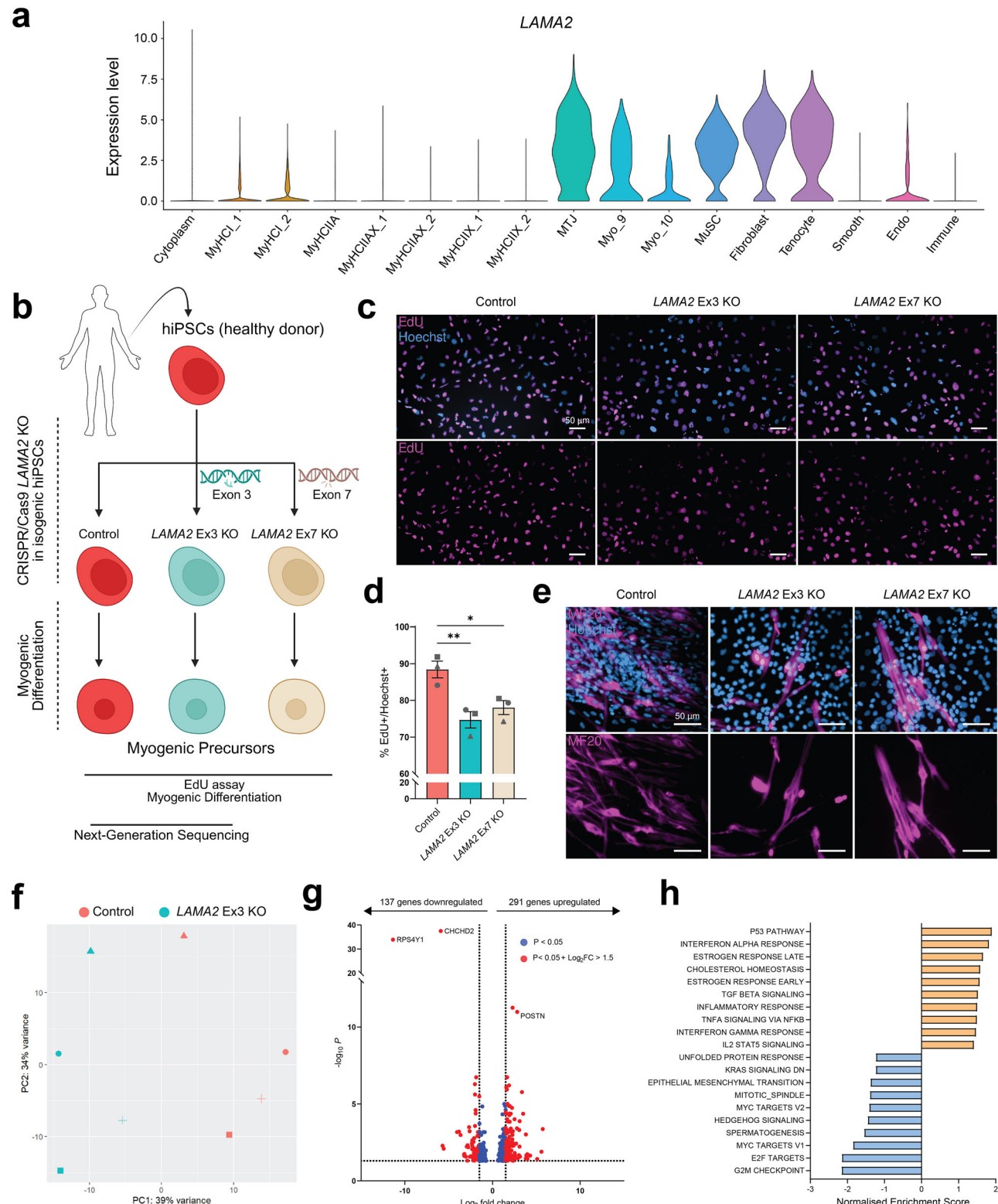

source, may play an essential role in supporting MuSC proliferation during the early stages of regeneration and, therefore, that its absence contributes to the impaired expansion of MuSCs in $dy^W/dy^W$ mice. However, after transplanting $dy^W/dy^W$ MuSCs into a wild-type environment and knocking out *Lama2* solely in MuSCs, we observed that laminin-α2 depletion in MuSCs was sufficient to slow MuSC proliferation. We interpret this as strong evidence that the abundance (Fig. S6m, S6o) and localization (Figs. 6h, S6k) of laminin-α2 in the early stages of regeneration rely on laminin-α2 secretion by MuSCs

themselves. Interestingly, while MuSC-specific *Lama2* knockout is sufficient to reduce the amount of laminin-α2 at 4 DPI (Fig. S6m, S6o), it does not influence the protein's abundance at 10 DPI (Fig. S6m, S6o). Hence, the MuSC-intrinsic requirement for laminin-α2 appears to affect only the early, proliferative phase of MuSCs, while later stages might be compensated for by other cell types. In line with this idea, we observed that by 10 DPI, MuSC-*Lama2*KO mice, like control mice, contained more fibers in the injured TA than the contralateral uninjured TA (Fig. 6g). This indicated that they generated new BM-

**Fig. 7 | *LAMA2* knockouts impair the proliferation of hiPSC-derived myogenic precursor cells. a** Violin plot showing *LAMA2* expression in snRNA-seq of human skeletal muscle. Nuclear identity was established by the authors[46]. Myo_9 and Myo_10 are myofiber clusters that were not assigned an identity due to a lack of clear marker expression[46]. **b** Experimental approach: hiPSCs were generated from a healthy donor. CRISPR/Cas9 was then employed to knockout *LAMA2* by excising exon 3[47] or introducing a frameshift mutation in exon 7. Control and *LAMA2* knockout hiPSCs were subsequently differentiated into myogenic precursors and characterized. **c** Representative immunostaining of control, *LAMA2* Ex3 KO and *LAMA2* Ex7 KO myogenic precursors after a 24 h incubation with EdU (EdU in magenta, Hoechst 33342 in blue). **d** Quantification of the proportion of EdU +/Hoechst+ cells after a 24 h incubation with EdU. Cells were differentiated in 3 replicate experiments (shown by symbol shapes), and for each experiment, 3 images were quantified per well for 2-3 separate wells. **e** Representative immunostaining of myotubes derived from control, *LAMA2* Ex3 KO and *LAMA2* Ex7 KO

hiPSCs (MF20 in magenta, Hoechst 33342 in blue). This experiment was performed once. **f** PC analysis of bulk RNA-sequencing data from control and *LAMA2* Ex3 KO myogenic precursors. Symbol shapes identify cells differentiated in parallel. **g** List of differentially-expressed genes in *LAMA2* Ex3 KO myogenic precursors. Blue: changes with a *P*-value < 0.05 and a Log2FC ≤ 1.5; red: changes with a *P*-value < 0.05 and a Log2FC change > 1.5. The number of upregulated and downregulated genes (*P*-value < 0.05) is shown above the graph. **h** Gene set enrichment analysis of the top-10 positively-enriched (orange) and negatively-enriched (blue) hallmarks in *LAMA2* Ex3 KO myogenic precursors. All enrichments have an FDR < 0.25. Data are means ± SEM. In **d**, statistical significance was determined by one-way ANOVA with Tukey's multiple comparisons test. In **g**, analysis was performed using DESeq2, with the Wald test for pairwise comparisons of expression levels. *$P < 0.05$; **$P < 0.01$. Source data and *P*-values are provided as a Source Data file. **b** Created in BioRender. Ruegg, M. (2025) https://BioRender.com/d18khcw.

surrounded muscle fibers in spite of the loss of laminin-α2 secretion by MuSCs and MuSC-derived myonuclei. In the future, inducible MuSC-specific *Lama2* knockouts could be extended to additional muscle-residing cells, such as FAPs or pericytes, to determine which cells contribute to the formation of new BMs during regeneration in the absence of laminin-α2 secretion by MuSCs. Additionally, severe injury models that eliminate residual BMs—such as freeze injuries[56]—could be employed to stimulate MuSC activation in an environment devoid of laminin-α2. This strategy would help distinguish the effects of MuSC-derived laminin-α2 from those of laminin-α2 retained in the BMs of ghost fibers. Our findings show that the residual laminin-α2, although it can guide the orientation of regenerating fibers[31,32], is not sufficient to support the timely expansion of MuSCs post-injury (Fig. 6e, f).

While laminin-α2 supports the proliferation of activated MuSCs, laminin-α2 per se does not systematically promote MuSC cell-cycle progression. As laminin-α2 is also highly abundant in the niche of non-proliferating, quiescent MuSCs (Figs. 1i, 5e), its influence on the cell-cycle must rely on more than just its presence or absence. A threshold of *Lama2* expression and laminin-α2 abundance, for example, could dictate whether it promotes MuSC proliferation or not. This hypothesis is supported by the blunted increase in laminin-α2 levels in the TA (Fig. S6m, S6o) and the coinciding delay in MuSC expansion (Fig. 6e, f) observed in MuSC-*Lama2*KO mice during the early stages of regeneration. An alternative hypothesis is that laminin-α2 relies on other ECM components to exert its pro-proliferative effects. This idea stems from observations that certain ECM proteins, such as fibronectin[57] and laminin-α5[18], are exclusively detected in the niche of activated MuSCs. Since the appearance of these proteins coincides with the initiation of laminin-α2's mitogenic influence, it is possible that the ECM composition modulates MuSC responsiveness to laminin-α2. In this manner, laminin-α2 would only promote MuSC proliferation when activation-specific ECM proteins are present. This coordinated response may be achieved directly by activating or repressing signaling pathways downstream of their receptors, or indirectly by modulating the stiffness of the MuSC niche[58]. Interestingly, since activated MuSCs express *Fn1*[57] and *Lama5* (Fig. S1h), the assembly of a permissive ECM microenvironment and thus the response to laminin-α2 could be regulated by the MuSCs themselves. Additionally, laminin-α2 could rely on specific growth factors to exert its mitogenic effect. This hypothesis is supported by the observation that in the central nervous system, laminin-α2 binds to receptors on oligodendrocytes and influences the cells' fate by amplifying and/or switching their response to neuregulin-1[59]. A similar mechanism, relying on growth factors that are present in the niche of activated MuSCs, could explain why laminin-α2 does not promote the proliferation of quiescent MuSCs. Finally, laminin-α2's influence on MuSC proliferation might be dictated by its localization. While laminin-α2 is restricted to the basal side of quiescent MuSCs (Fig. 1i), we show that activated MuSCs become surrounded by the protein as they secrete it

(Figs. 1g–i, 5e, 6h, S6k, Supplementary Movie 1). This loss of MuSC polarization, which is bolstered by the injury-induced elimination of apical interactions with the muscle fiber, may be essential for laminin-α2 to exert its pro-proliferative effects.

## Mechanisms regulating MuSC proliferation

Previous studies have shown that laminin-211, which contains laminin-α2, can bind to numerous receptors present on MuSCs, including the integrins α7β1 and α6β1[18,60,61]. We hypothesized that these receptors could mediate laminin-α2's influence on MuSC proliferation, as MuSCs lacking β1-integrin also exhibit proliferative defects post-injury[44]. Furthermore, both integrin α7β1 and α6β1 were shown to support the proliferation of neuronal progenitors in laminin-α2-rich environments of the developing brain[62]. We tested the activation state of β1-integrin-containing receptors with a specific antibody (9EG7) and detected a trend towards lower activation in $dy^W/dy^W$ PMs, but not MuSC-*Lama2*KO PMs (Fig. S8c, S8d). Activation of β1-integrin-containing receptors on MuSCs in the absence of laminin-α2 is likely based on their engagement with other ligands, such as fibronectin, which activates integrin α5β1[45]. Nevertheless, since the phosphorylation of β1-integrin's downstream signaling effectors was also reduced in *Lama2*-deficient cells (Fig. S8a, S8b), these results suggest a contribution of β1-integrins but also imply that self-secreted laminin-α2 could support MuSC proliferation via additional receptors, such as α-dystroglycan[63]. Moreover, laminin-α2 may exert its effects through receptor-independent mechanisms, for example, by modulating the stiffness of the niche[58] or by influencing the diffusion of signaling molecules[64]. Our transcriptomic analysis of human myogenic precursors highlighted that the *LAMA2* knockout alters several pathways: genes related to Hedgehog and MYC signaling were downregulated, while genes linked to the p53 pathway were enriched in *LAMA2* Ex3 KO cells (Fig. 7h). These pathways have all been implicated in the regulation of MuSC proliferation[65–67] and may offer further insights into the mechanisms underlying laminin-α2's influence on the cell-cycle. However, their modulation could also be a secondary effect of broader signaling changes.

## Implications for LAMA2 MD

Current therapeutic strategies under development for LAMA2 MD include gene therapies aimed at replacing *Lama2*, either by CRISPR-dCas9-mediated upregulation of *Lama1*[68] or by the delivery of "linker proteins"[4,69]. These approaches have been shown to reduce muscle fiber frailty, fibrosis, and inflammation, leading to a marked amelioration of the disease phenotype in *Lama2*-deficient mouse models. Similar to recent efforts to enhance MuSC function in LAMA2 MD[8], these approaches might indirectly improve regeneration by altering the cells' dystrophic environment. Here, we show that simply removing these cell-extrinsic factors by transplanting $dy^W/dy^W$ MuSCs into non-dystrophic mice is insufficient to fully restore their regenerative potential. Beyond the persistent cell-intrinsic defects caused by *Lama2* mutations,

transplanted $dy^W/dy^W$ MuSCs may also carry enduring "scars" imprinted by their original dystrophic microenvironment. For instance, chronic exposure to inflammation could permanently impair their function by inducing telomere shortening[23]. Repeated cycles of degeneration/regeneration may also alter their epigenetic profile[70], further compromising their regenerative capacity even after transplantation. This combination of extrinsic and intrinsic factors likely underlies the more severe regenerative deficiencies observed in $dy^W/dy^W$ mice compared to MuSC-*Lama2*KO mice. It may also explain why strategies such as increasing endothelial cell numbers[8], enhancing muscle fiber stability or inhibiting apoptosis[71] can still improve muscle regeneration in $dy^W/dy^W$ mice, despite the persistence of cell-intrinsic defects.

In summary, we demonstrate that loss-of-function mutations in *Lama2* impair MuSC proliferation and compromise muscle regeneration. Our findings uncover a previously unknown, cell-intrinsic role for laminin-α2 in MuSCs that allows efficient muscle regeneration. Collectively, these results advance our understanding of LAMA2 MD and identify MuSCs as additional therapeutic targets. While further experiments are needed to determine whether improving MuSC function alone is sufficient to ameliorate the disease, a multifaceted therapeutic approach that addresses the muscle fiber frailty as well as the extrinsic and intrinsic disruptions of MuSC function is likely to yield the most beneficial outcomes to combat muscle wasting in this severe congenital muscular dystrophy.

# Methods

## Mice
All mice were kept on a 12 h light–dark cycle (6 AM–6 PM) at 22 °C (range 20–24 °C) and 55% (range 45–65%) relative humidity. For all experiments, the dystrophic mice were compared to healthy wild-type littermates. Long-necked water bottles and wet food were given to all cages to ensure dystrophic mice can access food and water.

$dy^W/dy^W$ mice (B6.129S1(Cg)-Lama2<tm1Eeng>; Jackson Laboratory stock #013786) containing a LacZ insertion in the *Lama2* gene[72] were the main LAMA2 MD mouse models used in the study. $dy^{3K}/dy^{3K}$ mice (B6.129P2(Cg)-*Lama2*<tm1Stk>, a kind gift from Drs. Shin'ichi Takeda and Yuko Miyagoe-Suzuki), which have a total absence of laminin-α2[73], were also used to characterize uninjured tissue (Fig. S3a–e). For transplantation experiments (Figs. 4, S4, S5), wild-type and $dy^W/dy^W$ mice with an inducible EGFP reporter in Pax7+ MuSCs were generated by crossing heterozygous $dy^W$/wt mice with Pax7<sup>creER</sup> mice[29] (Jackson Laboratory strain #017763) and mice with a Cre-inducible CAG-EGFP inserted into the *Rosa 26* locus[30] (B6.Cg-Pax7<tm1(cre/ERT2)Gaka> x B6.C-Gt(ROSA)26Sor<tm8(CAG-EGFP)Npa> x B6.129S1(Cg)-Lama2<tm1Eeng>). NOD *scid* gamma mice (NSG) (NOD.Cg-Prkdc<scid>Il2rg<tm1Wjl>SzJ, Jackson Laboratory strain #005557) were used as recipient mice during the transplantation experiments.

To knock out *Lama2* in MuSCs, two mouse models were used. Mice with inducible Cas9 and EGFP expression in Pax7+ cells (Fig. S7) were generated by crossing Rosa26-LSL-Cas9 knock-in mice[41] (Jackson Laboratory Strain, #026175) with Pax7<sup>creER</sup> mice[29] (Jackson Laboratory strain #017763) (B6.129S6(Cg)-Pax7<tm1(cre/ERT2) Gaka> x B6J.129(B6N)-Gt(ROSA)26Sor<tm1(CAG-cas9*,-EGFP)Fezh>/ J). Additionally, *Lama2*-floxed mice (C57BL/6J Lama2<em1(fl)Rueg>) were generated in collaboration with the Basel Center for Transgenic Models by inserting loxP sites into the introns flanking exon 3 of *Lama2*. They were then crossed with Pax7<sup>creER</sup> mice[29] (Jackson Laboratory strain #017763) and mice with a Cre-inducible cytosolic CAG-EGFP inserted into the *Rosa 26* locus[30] for MuSC-specific *Lama2* knockout and EGFP labeling.

## Cardiotoxin injury
To induce injuries, mice were anaesthetized with isoflurane and injected in the TA muscle with 50 μl of 10 μM cardiotoxin (Latoxan) in 0.9% NaCl (Fresenius Kabi).

## Fluorescence-activated cell sorting (FACS)
Hindlimb and forelimb muscles were collected in 500 μl of Ham's F-10 media (Gibco) and minced using scissors. Minced muscles were then transferred to a GentleMACS C tube (Miltenyi) and 5 ml of pre-warmed digestion buffer (1% Collagenase B (Roche), 0.4% Dispase II (Roche), 25 mM HEPES (Bioconcept) in Ham's F-10 media (Gibco)) was added. Tubes were then mounted on a GentleMACS Octo Separator (Miltenyi) and the following program was run: Temp ON 37 °C, 90 rpm 5 min, −90 rpm 5 min, loop 10x (360 rpm 5 s, −360 rpm 5 s), end loop, 90 rpm 5 min, −90 rpm 5 min, end. 5 ml of FBS (Biological Industries) was then added to halt the digestion, and the content of the tube was transferred to a 50 ml conical tube through a 70 μm filter, which was subsequently washed using 20 ml of cold FACS buffer (2% FBS (Biological Industries), 2.5 mM EDTA in PBS). The samples were then centrifuged for 5 min at 500×*g* and 4 °C and resuspended in 500 μl of cold FACS buffer. 2 ml of red blood cell lysing buffer (Sigma) was added for 5 min at room temperature (RT), the lysates were then diluted in 10 ml of cold FACS buffer and centrifuged for 5 min at 500×*g* and 4 °C. The cells were then resuspended in FACS buffer and stained with antibodies (Supplementary Data 1) for 10 min at 4 °C. After the staining, 10 ml of cold FACS buffer was added to the cells, and the tubes were centrifuged at 500×*g* and 4 °C for 5 min. Finally, the cells were resuspended in cold FACS buffer, strained through a 40 μm filter and sorted using BD FACSAria™ III and FACSAria™ Fusion sorters (BD Biosciences). DAPI (Thermo Fisher Scientific) was added to the samples immediately before sorting to remove dead cells. Gating boundaries were determined by single-stain and fluorescence minus one controls.

## MuSC transplantation
For EGFP labeling of MuSCs, 4-week-old wild-type and $dy^W/dy^W$ Pax7<sup>CreERT2/</sup>/R26R<sup>EGFP/+</sup> mice were injected intraperitoneally with tamoxifen at doses of 75 mg/kg once daily for 5 consecutive days. To enable MuSC engraftment into the TA upon transplantation, 8-week-old female NSG mice were injured using cardiotoxin (see Cardiotoxin Injury). One day later, muscles of wild-type and $dy^W/dy^W$ Pax7<sup>CreERT2/wt</sup>/ R26R<sup>EGFP/wt</sup> mice were harvested, and EGFP+ MuSCs were isolated via FACS. Cells were collected in FACS buffer, centrifuged for 5 min at 500×*g* and 4 °C, and resuspended to a concentration of 1000 cells per μl in 0.9% NaCl (Fresenius Kabi). NSG mice were then anaesthetized using isoflurane, and 10,000 cells were injected into their injured TA using a 28 G needle (30° angle, tip type 4, Hamilton) with a 10 μl Hamilton syringe (Hamilton). Cells isolated from one donor mouse were transplanted into a single recipient mouse.

Mice collected at 3 days post-injury were injected intraperitoneally with 100 mg/kg 5-ethynyl-2′-deoxyuridine (EdU) (Invitrogen) 3 h before tissue collection. The mice were then sacrificed using a lethal dose of pentobarbital and transcardially perfused with 4% PFA for 10 min.

## Murine primary myoblast culture
Sorted cells were collected in proliferation medium (20% Fetal Bovine Serum (Biological Industries), 2.5 ng/ml basic Fibroblast Growth Factor (Gibco), 1% Penicillin/Streptomycin in DMEM (Gibco)) and plated on collagen (Sigma)-coated tissue culture dishes for expansion. All cells were incubated at 37 °C in 5% CO₂. To quantify the proportion of cells that replicate their DNA, 10 μM EdU (Invitrogen) was added to the proliferation medium either immediately post-sorting or after the fourth passage. For daily imaging, cells were cultured in proliferation medium containing clear DMEM (Gibco) and imaged daily on a brightfield microscope (Olympus). Three images were taken per well to calculate the average cell density for 5 days post-plating. Flow cytometry-mediated cell-cycle analysis[34] (Fig. 3d) was performed by fixing cells maintained in proliferation medium (passaged 4 times) in 70% ethanol for 2 h at 4 °C, washing the cells 3 times with PBS, equalizing their concentration and staining them with 1 μg/ml DAPI (Thermo

Fisher Scientific) in 0.1% Triton-X-100 in PBS for 30 min at RT. The cells were then washed with PBS, and the DAPI signal was measured on a BD FACSAria™ Fusion (BD Biosciences). Cell-cycle profiling was performed with FlowJo™ v10.8 Software (BD Life Sciences) using the Watson Pragmatic model[74]. MuSC differentiation was induced by removing proliferation medium, washing once with DMEM (Gibco), and adding differentiation medium (2% Horse Serum (Biological Industries), 1% Penicillin/Streptomycin in DMEM (Gibco)). For immunostaining, the cells were maintained in differentiation medium for 7 (Fig. 3g) or 3 days (Fig. S6c) (see Immunostaining of murine primary myoblasts for more details). For RT-qPCR, isogenic cells were cultured in two separate tissue culture dishes: in one, they were maintained in proliferation medium, and in the other, they were cultured in differentiation medium for 24 h. RNA was then extracted from cells in both dishes at the same time (see RNA extraction, isolation and RT-qPCR). For AAV-mediated transduction of Cas9-expressing cells ex vivo (Fig. S7), cells were plated in 35 mm collagen-coated tissue culture dishes, and AAVs were added to the medium at a titer of $1 \times 10^6$ vector genomes per cell. Four days later, transduced EGFP+/tdTomato+ cells were sorted via FACS and plated on collagen-coated tissue culture dishes for analyses.

## Single fiber isolation and culture
EDL or GAS muscles were harvested from both legs and directly placed in 0.2% Collagenase Type I (Sigma) in isolation medium (DMEM (Gibco) with 1% Penicillin/Streptomycin at 37 °C. After 45 min of digestion, the muscles were transferred to warm isolation medium in a dish pre-coated with 20% horse serum (Biological Industries). The muscles were then flushed using a wide-bore glass pipette to dissociate single fibers. These steps were performed at room temperature for a maximum of 5 min before the dishes were placed in a 37 °C incubator for 30 min to allow medium re-equilibration. Once a sufficient number of fibers were obtained, they were individually transferred to dishes containing culture media (20% FBS (PAN-Biotech), 1% Chicken Embryo Extract (Mpbio) and 2.5 ng/ml bFGF (Gibco) in isolation media) and incubated at 37 °C and 5% $CO_2$ until collection for immunostaining (see Immunostaining of single fibers).

## Generation and differentiation of human induced pluripotent stem cell lines
The generation of the DMBi001-A hiPSC line[75] and the derivation of the *LAMA2* Ex3 KO line were previously described in detail[47]. The *LAMA2* Ex7 KO line was generated using a similar methodology to that described for the *LAMA2* Ex3 KO line[47]. Briefly, a pSpCas9(BB)-2A-Puro plasmid (Addgene #62988) into which sgRNAs targeting *LAMA2* exon 7 (Supplementary Table 1) were cloned was introduced into control hiPSCs using the Human Stem Cell Nucleofector kit 1 (Lonza) and Nucleofector 2b Device (Lonza) according to the manufacturer's protocol. After nucleofection, hiPSCs were seeded on Geltrex™ LDEV-Free Reduced Growth Factor Basement Membrane (Thermo Fisher Scientific) coated plates in Essential 8 (E8) medium (Gibco) supplemented with 10 μM Y27632-ROCK kinase inhibitor (Abcam). 24 h later, the medium was replaced with fresh E8 supplemented with 0.5 μg/ml puromycin (Sigma) for the selection of nucleofected cells. After 24 h, the selection medium was removed and fresh E8 was added. Cells were then counted and seeded on Geltrex™-coated 10 cm plates at a density of 500 cells/plate to generate single cell-derived clones. Once clones could be picked and further expanded, DNA was isolated using the Genomic Mini kit (A&A Biotechnology) and Surveyor Nuclease Assay (SNA)[76] was performed to detect genetic modifications introduced by the sgRNAs in the desired locus. Clones harboring indels in *LAMA2* exon 7 were further evaluated by Sanger sequencing of the targeted genomic region, and a successfully-edited *LAMA2* Ex7 KO clone was selected for further characterization. RT-PCR was used to confirm the loss of *LAMA2*

mRNA in both *LAMA2* Ex3 KO and *LAMA2* Ex7 KO hiPSCs (Fig. S10) (see RNA extraction, isolation and RT-PCR/RT-qPCR).

For myogenic differentiation, hiPSCs were differentiated with the Skeletal Muscle Differentiation Kit (Amsbio) following the manufacturer's protocol with minor adjustments. Briefly, 17,000 hiPSCs per well were seeded on Geltrex™-coated 24-well plates in skeletal muscle induction medium and cultured for 8 days with medium change every other day. Subsequently, cells were harvested in TrypLE™ Express Enzyme (Thermo Fisher Scientific), centrifuged at 200×*g* for 5 min, counted and reseeded in skeletal myoblast medium. For the EdU incorporation assay, 10,000 myogenic precursor cells were seeded in Geltrex™-coated 48-well plates, and 10 μM EdU (Invitrogen) was added to the medium for 24 h. Cells were then fixed and EdU was detected according to the manufacturer's protocol with 1 μg/ml Hoechst 33342 nuclear staining. For further differentiation of myogenic precursors into myotubes, 17,000 myogenic precursors (day 8 of the differentiation protocol) were seeded on Geltrex™-coated 24-well plates in skeletal myoblast medium. The medium was changed every other day until day 16, when it was replaced by myotube fusion medium. Cells were cultured for another 8 days with medium change every third day until reaching proper myotube morphology.

## RNA-sequencing of human myogenic precursors
For RNA-sequencing, RNA was isolated from control and *LAMA2* Ex3 KO myogenic precursors by transferring and resuspending cells in 400 μl Fenozol reagent (A&A Biotechnology). 100 μl of chloroform was added to the samples, which were then vortexed, incubated on ice for 20 min and centrifuged for 20 min at 10,000×*g* and 4 °C. The aqueous phase was collected, and an equal amount of isopropanol was added. Samples were incubated overnight at −20 °C before centrifugation for 30 min at 10,000×*g* and 4 °C. Then, pellets were washed twice with 70% ethanol and centrifuged for 10 min at 10,000×*g* and 4 °C before resuspension in ddH$_2$O. RNA concentration was measured with a NanoDrop (Thermo Fisher Scientific). RNA Libraries were prepared with the Ion AmpliSeq Transcriptome Human Gene Expression Kit (Thermo Fisher Scientific) and sequenced on an Ion Proton™ Sequencer (Thermo Fisher Scientific) using the Ion PI Hi-Q Sequencing 200 Kit (Thermo Fisher Scientific) and the Ion PI™ Chip Kit v3 (Thermo Fisher Scientific).

Differential gene expression analysis was performed using the DESeq2[77] package from Bioconductor on R (version 4.1.2), with donor identity included as a covariate in the design formula to control for inter-individual variability. Genes with low expression (total counts ≤ 1 across all samples) were filtered out to reduce background noise. A DESeqDataSet object was constructed using the filtered count matrix and metadata, with the design formula - Exp + Group. To stabilize variance across the range of mean expression values, a variance stabilizing transformation (VST) was applied, and VST-transformed data were used for exploratory analyses. Sample relationships were visualized using heatmaps based on Euclidean and Poisson distances, and principal component analysis (PCA) was conducted to assess clustering by experimental condition. Differential expression testing was carried out using the DESeq2 pipeline, and results were filtered based on adjusted *P*-values (FDR < 0.1) and log2-fold change thresholds. Shrinkage of log2-fold changes was performed using the apeglm method to improve interpretability. Annotated results were generated using the org.Hs.eg.db package to map gene symbols, Entrez IDs, and Ensembl IDs, and the top 100 genes were exported for further analysis. Gene set enrichment analysis (GSEA)[52] was performed using the fgsea package, with genes ranked by log2-fold change and tested against hallmark gene sets from MSigDB.

## Immunostaining
**Muscle cryo-sections.** Following dissection, muscles were embedded in optimal cutting temperature (O.C.T., Tissue-Tek) and frozen in

liquid nitrogen-cooled isopentane for 20 s before storage at −80 °C. For EGFP-containing muscles, dissected muscles were immediately placed in 4% paraformaldehyde (PFA) for 2 h at 4 °C, then in 20% sucrose overnight at 4 °C before embedding and freezing as described above. 10 or 15 μm sections were cut at −20 °C on a cryostat (Leica, CM1950) and collected on SuperFrost Plus (VWR) slides. The sections were fixed in 4% PFA for 10 min at room temperature, washed in PBS, and blocked in 10% goat serum (Biological Industries) in 0.1% Tween-20 in PBS (PBST). If a mouse or rat primary antibody was used, 1:40 mouse-on-mouse reagent (Vector Labs) was added to the blocking solution. Sections were then incubated in primary antibody solution (Supplementary Data 1) overnight at 4 °C, washed 4 times for 5 min in PBS, incubated in a secondary antibody solution (Supplementary Data 1) for 1 h at RT, washed 4 times for 5 min in PBS and mounted using ProLong™ Gold antifade with DAPI (Invitrogen). EdU (Invitrogen) detection was performed according to the manufacturer's protocol after the secondary antibody incubation. It was followed by 4 washes in PBS for 5 min before mounting in ProLong™ Gold antifade with DAPI (Invitrogen). Terminal deoxynucleotidyl transferase dUTP nick-end labeling (TUNEL) staining was performed according to the manufacturer's protocol for the Click-iT™ Plus TUNEL Assay (Invitrogen) after fixation and before blocking. The rest of the immunostaining protocol was performed as described above. Images were taken in the Biozentrum Imaging Core Facility using a SpinD confocal microscope (Olympus) or a Zeiss Axio Scan.Z1 (Zeiss).

**Murine primary myoblasts.** Murine PMs were washed with PBS supplemented with $Ca^{2+}$ and $Mg^{2+}$ and fixed for 10 min at RT in 4% PFA. They were then washed 3 times for 5 min in 0.1 M glycine in PBS (pH 7.4). After the final wash, cells were permeabilized in 0.2% Triton X-100 in PBS for 10 min and blocked for a minimum of 1 h at RT in 5% bovine serum albumin (Sigma-Aldrich) in PBS. Primary antibodies (Supplementary Data 1) were added for an overnight incubation at 4 °C. The cells were then washed 3 times for 5 min with PBS before incubation for 1 h at RT in secondary antibody solutions (Supplementary Data 1). Finally, cells were washed 3 times for 5 min in PBS and mounted in ProLong™ Gold antifade with DAPI (Invitrogen). EdU (Invitrogen) detection was performed according to the manufacturer's protocol after the secondary antibody incubation. It was followed by 4 washes in PBS for 5 min before mounting in ProLong™ Gold antifade with DAPI (Invitrogen). Images were taken in the Biozentrum Imaging Core Facility using a SpinD confocal microscope (Olympus). For calculations of Corrected Total Cell Fluorescence (CTCF), the following equation was used: CTCF = Integrated density−(Area of selected cell × Mean fluorescence of background readings).

**Human myogenic precursor cells.** For the staining of human myogenic precursors, cells were washed with PBS without $Ca^{2+}$ and $Mg^{2+}$ and fixed for 15 min at RT in 4% PFA. They were then washed 3 times for 5 min in PBS. After the final wash, cells were permeabilized in 0.1% Triton X-100 in PBS for 15 min and blocked for a minimum of 1 h at RT in 5% Bovine Serum Albumin (Sigma-Aldrich) in PBS. Subsequent steps were identical to those described for murine PMs, with the exception of nuclear staining, which was performed by adding 1 μg/ml Hoechst 33342 to the secondary antibody solution. EdU (Invitrogen) detection was performed according to the manufacturer's protocol after the secondary antibody incubation. It was followed by 4 washes in PBS for 5 min before mounting in ProLong™ Gold antifade (Invitrogen). Images were taken on a Nikon Eclipse Ti fluorescent microscope (Nikon).

**Single fibers.** Fibers were transferred into an Eppendorf pre-coated with 20% horse serum (Biological Industries), washed once with PBS for 5 min, fixed in 4% PFA for 10 min and washed 3 times in 0.1 M glycine in PBS (pH 7.4). Fibers were then permeabilized with 0.2% Triton X-100 in PBS for 10 min at RT before blocking with 10% goat

serum (Biological Industries) in PBS for a minimum of 1 h at RT. The samples were then incubated in primary antibody solutions (Supplementary Data 1) overnight at 4 °C, washed 3 times in PBS, incubated in a secondary antibody solution (Supplementary Data 1) for 1 h at RT, washed 3 times in PBS and mounted on glass slides with ProLong™ Gold antifade with DAPI (Invitrogen). Images were taken in the Biozentrum Imaging Core Facility using a SpinD confocal microscope (Olympus).

**Whole-mounts.** TA or EDL muscles were dissected, pinned on a sylgard dish, fixed with 4% PFA in PBS for 10 min at RT, washed 3 times in 0.1 M glycine in PBS (pH 7.4) and separated into small bundles using tweezers. The bundles were then permeabilized with 0.2% Triton X-100 in PBS for 30 min, washed twice with PBS for 5 min and blocked in 10% goat serum (Biological Industries) in PBS for a minimum of 3 h at RT. The samples were then incubated in primary antibody solutions (Supplementary Data 1) overnight at 4 °C, washed 4 times in PBS, incubated in a secondary antibody solution (Supplementary Data 1) for 1.5 h at RT, washed 4 times in PBS and mounted on glass slides with ProLong™ Gold antifade with DAPI (Invitrogen). EdU (Invitrogen) detection was performed according to the manufacturer's protocol after the secondary antibody incubation. It was followed by 4 washes in PBS for 10 min before mounting in ProLong™ Gold antifade with DAPI (Invitrogen). Images were taken in the Biozentrum Imaging Core Facility using a SpinD confocal microscope (Olympus).

### Inorganic stainings
H&E staining (Merck) and Picro Sirius Red stains (Direct Red 80 (Sigma) in picric acid solution) were performed for histological analyses. Images were taken with an Axio Scan.Z1 Slide Scanner (Zeiss) on the Zen 3.5 software. The proportion of tissue stained by Sirius Red was quantified by measuring the area of Sirius Red staining and dividing it by the total area of the muscle cross-section.

### RNA extraction, isolation and RT-PCR/RT-qPCR
In Fig. 1b and S8e, RNA was extracted from murine cells using the RNeasy Fibrous Tissue kit (QIAGEN). RNA yield and quality were determined with a NanoDrop 1000 spectrophotometer (Thermo Fisher Scientific). cDNA was prepared with the iScript cDNA synthesis kit (Bio-Rad). FastStart Universal SYBR Green Mastermix (Roche) was used for real-time PCR with the StepOnePlus Real-Time PCR System (Applied Biosystems). The list of primers can be found in Supplementary Table 2. Quantifications were performed using the $2^{-\Delta ct}$ method.

In Fig. S10, RNA was extracted from hiPSCs as previously described (see RNA-sequencing of human myogenic precursors). RNA yield and quality were determined with a NanoDrop 1000 spectrophotometer (Thermo Fisher Scientific). Two micrograms of RNA were then reverse-transcribed to cDNA using the High-Capacity RNA-to-cDNA™ Kit (Applied Biosystems), following the manufacturer's instructions, in a ProFlex PCR System (Thermo Fisher Scientific). PCR was performed using 50 ng of cDNA, gene-specific primers (Supplementary Table 2), and KAPA2G Fast Genotyping Mix (Merck) in the same thermocycler. Amplification products were resolved by electrophoresis on a 2% agarose gel. DNA Marker 1 (A&A Biotechnology) was used to estimate product sizes.

### Single-molecule RNA fluorescent in situ hybridization (smRNA FISH)
For muscle cross-sections, tissues were prepared as described above (see Immunostaining muscle cryo-sections). For cells in culture, PMs were cultured in the chambers of an 8-well slide (Ibidi). The following protocol was followed for both cells and muscle cross-sections. Slides were fixed for 30 min in 4% PFA at 4 °C, washed twice in PBS for 5 min and serially dehydrated for 5 min in 50%, 70% and twice in 100%

ethanol. Slides were then left to dry at RT, and the tissues or cells were circled with a Hydrophobic Barrier Pen (Vector Laboratories). For Figs. 1a, S1b and S9c, protease IV was added for 30 min at RT, before 3 washes in ddH₂O for 2 min each. For Figs. 1f and S1f, protease IV treatment was omitted so that staining of Pax7, Myogenin and Ki67 could be performed subsequently. mRNA hybridization against *Pax7* (314181; Figs. 1a, S1b, S9c), *Myog* (492921; Figs. 1a, S9c), *Lama2* (424661; Figs. 1a, f, S1b, S1f) and *Lama4* (494901; Fig. S9c) and all subsequent steps were performed according to the manufacturer's instructions for the Multiplex V1 assay (Figs. 1a, f, S1b, S1f) or V2 assay (Fig. S9c) (Advanced Cell Diagnostics) at 40 °C in a HybEZ™ oven (Advanced Cell Diagnostics). For Figs. 1a, S1b and Fig. S9c, slides were mounted using ProLong™ Gold antifade with DAPI (Invitrogen) at the end of the hybridization protocol. For Figs. 1f and S1f, immunostaining was performed after the smRNA FISH protocol. Sections were blocked for 1 h in 10% goat serum (Biological Industries) in PBST, incubated overnight at 4 °C with Pax7 and Myogenin primary antibodies (Supplementary Data 1) (Fig. 1f) or Pax7 and Ki67 primary antibodies (Supplementary Data 1) (Fig. S1f) in 10% goat serum (Biological Industries) in PBS. They were then washed four times for 5 min in PBS, incubated for 1 h at RT with secondary antibodies (Supplementary Data 1) in 10% goat serum (Biological Industries) in PBS, washed four times for 5 min in PBS and mounted using ProLong™ Gold antifade with DAPI (Invitrogen). Images were taken in the Biozentrum Imaging Core Facility using a SpinD confocal microscope (Olympus).

## Protein isolation and Western blot analysis

Snap-frozen TAs were pulverized on a metal plate chilled in liquid nitrogen and lysed in ice-cold RIPA buffer (50 mM Tris–HCl pH 8.0, 150 mM NaCl, 1% NP-40; 0.5% deoxycholate; 0.1% SDS; 20 mM EDTA) supplemented with protease and phosphatase inhibitors (Roche). Lysates were then sonicated twice for 10 s, incubated on a rotating wheel for 2 h at 4 °C and centrifuged at 16,000×*g* for 20 min at 4 °C. Supernatants were collected, and protein concentrations were equalized after measurement with the Pierce BCA Protein Assay Kit (Thermo Fisher Scientific) according to the manufacturer's protocol. For protein extraction from cells cultured in 35 mm collagen-coated dishes ex vivo (Fig. S8a), the culture medium was aspirated, and cells were rapidly washed with cold PBS. 300 μl cold RIPA buffer (50 mM Tris–HCl pH 8.0, 150 mM NaCl, 1% NP-40; 0.5% deoxycholate; 0.1% SDS; 20 mM EDTA) was added to the cells for a 5 min of incubation on ice, after which cells were scraped and collected in RIPA buffer, incubated on a rotating wheel for 2 h at 4 °C and centrifuged at 16,000×*g* for 20 min at 4 °C. Supernatants were collected, and protein concentrations were equalized after measurement with the Pierce BCA Protein Assay Kit (Thermo Fisher Scientific) according to the manufacturer's protocol. For the detection of laminin-α2 (Fig. S2a, S6l), proteins were separated on a 3–8% Tris–acetate gel (Invitrogen) in NuPAGE Tris–Acetate SDS running buffer (Invitrogen) before transfer with the iBlot 3 Western Blot Transfer System (Thermo Fisher Scientific) High Molecular Weight program. For the detection of p-Erk1/2, p-Akt, total Erk1/2 and total Akt (Fig. S8a), proteins were separated on a 4-12% Bis-Tris protein gel (Invitrogen) in NuPAGE MOPS SDS running buffer (Thermo Fisher Scientific) before transfer with the iBlot 3 Western Blot Transfer System (Thermo Fisher Scientific) Low Molecular Weight program. For all experiments, membranes were then stained with Ponceau S solution (Sigma-Aldrich) and imaged on a Bio-Rad ChemiDoc MP Imaging System (Bio-Rad). Membranes were blocked for 3 h in 5% BSA in PBST (0.1% Tween-20 in PBS) at RT before incubation with primary antibodies (Supplementary Data 1) overnight at 4 °C. Following 3 washes for 10 min each in PBST, the membranes were incubated with secondary antibodies (Supplementary Data 1) for 1 h at RT. After 2 washes in PBST and one wash in PBS, proteins were detected using a SuperSignal™ West Femto Maximum Sensitivity Substrate kit (Thermo Fisher Scientific) (Figs. S2a, S6l) or a Lumiglo Chemiluminescent Substrate

System (Seracare) (Fig. S8a) and imaged on a Bio-Rad ChemiDoc MP Imaging System (Bio-Rad). To calculate the ratios of p-Erk1/2 to total Erk1/2 and p-Akt to total Akt (Fig. S8a, S8b), p-Erk1/2 and p-Akt were first detected as described above. After their detection, the membranes were placed in PBS for 10 min at RT and stripped with Restore Western Blot Stripping Buffer (Thermo Fisher Scientific) for 25 min at RT. Membranes were then washed for 10 min in PBS at RT and blocked in 5% BSA in PBST for 1 h at RT. All subsequent steps to detect total Erk1/2 and total Akt were identical to those described above for p-Erk1/2 and p-Akt. Quantifications were performed using the Image Lab 6.1 Software (Bio-Rad).

## Design of single guide RNAs and production of AAVs

sgRNAs (Supplementary Table 1) were designed using CRISPOR v5.01[78]. The 3 sgRNAs were cloned between AAV serotype 2 ITR's including a cloning site for multiplexed hU6-sgRNA insertions (MluI and KpnI (NEB)), the ubiquitous CMV promoter, tdTomato, WPRE and bovine growth hormone polyA signal. For AAV production, HEK293T cells (CRL-3216, ATCC) were transfected with transfer plasmid (AAV-tdTomato construct or AAV-3xsgRNA-tdTomato), AAV helper[42] and pAd-DeltaF6 helper plasmid (a gift from J.M. Wilson, Addgene, plasmid # 112867) using PEI MAX (Polyscience). Ten 15-cm tissue culture plates were processed for each preparation. The supernatant was collected at 48 and 72 h post-transfection. Cells were then dislodged at 72 h post-transfection with PBS, before being centrifuged at 500×*g* at 4 °C for 10 min and resuspended in AAV lysis solution (50 mM Tris–HCl, 1 M NaCl, 10 mM MgCl₂, pH 8.5). 1000 U of salt active nuclease (Sigma), which were added to the cells that were then incubated at 37 °C for 1 h with continuous shaking. Lysates were centrifuged at 4000×*g* for 15 min at 4 °C. The supernatant was collected for AAV precipitation. Polyethylene glycol 8000 (Sigma) was added to the supernatant to a final concentration of 8% (w/v). Samples were then incubated for 2 h at 4 °C before centrifugation at 4000×*g* for 30 min at 4 °C. The pellet was resuspended in AAV lysis buffer and pooled with the cell lysate. A 15–25–40–60% iodixanol (Serumwerk) gradient was used to purify AAV particles. The gradient was centrifuged at 292,000×*g* (Beckman type 70 Ti rotor) for 2 h at 4 °C, and the AAV particles were collected from the 40–60% phase interface. The extract was filtered through a 100 kDa MWCO filter (Millipore) and washed with PBS containing 0.01% Pluronic F-68 surfactant (Gibco) until the buffer exchange was complete. AAVs' titers were measured by digital PCR with the ITR2/5 assay (Qiagen).

## Genomic DNA isolation, PCR amplification and TIDE

Genomic DNA was extracted from cells in culture using the DNeasy blood and tissue kit (Qiagen) according to the manufacturer's protocol. To assess exon 3 removal in MuSC-*Lama2*KO cells (Fig. 5b), DNA was amplified using LongAmp Taq polymerase (NEB). Primers for this PCR are available in Supplementary Table 2. For characterization of CRISPR-editing efficiency (Fig. S7b), DNA was amplified using LongAmp Taq polymerase (NEB) targeting the sgRNA editing sites with 200–500 bp-long amplicons. The primers used for this are described in Supplementary Table 2. PCR amplicons were purified using AMPure XP beads (Beckman) and Sanger-sequenced (Microsynth) using one of the two PCR primers. Sequencing chromatograms were then analyzed with TIDE[79] to quantify CRISPR-editing efficiency of the target locus.

## Statistics and reproducibility

All values are expressed as mean ± SEM, unless stated otherwise. All statistical methods used to analyze the data are described in the Figure legends and were performed using GraphPad Prism 9 software. Statistical significance was set at a *P*-value < 0.05. No statistical method was used to predetermine sample size. No data were excluded from the analyses.

## Ethical statement

All procedures involving animals were performed in accordance with Swiss regulations and approved by the veterinary commission of the canton Basel-Stadt.

## Data availability

Requests for further information and resources should be directed to and will be fulfilled by the lead contact, Markus A. Rüegg (markus-a.ruegg@unibas.ch). Sequencing data generated in this study is deposited in GEO under the accession number: GSE307443. The previously described script for automated muscle cross-section analysis (Myosoft)[80] was further developed in-house and is available on GitLab (https://git.scicore.unibas.ch/imcf/myosoft-imcf). Source data are provided with this paper.

## Code availability

The code used for bulk RNA-sequencing analysis is publicly available from Zenodo [https://doi.org/10.5281/zenodo.17310433].

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

## Acknowledgements

We thank the Biozentrum core facilities for their technical support with imaging (Imaging Core Facility), cell sorting (FACS Core Facility) and housing of the mice (Animal Facility). We also thank Jesús Dante Ramírez López for his technical support with hiPSCs. T.J.M. was supported by a Biozentrum Ph.D. Fellowship. M.A.R. and J.R.R. were supported by the Muscular Dystrophy Association (grant: https://doi.org/10.55762/pc.gr.157038). The remaining support awarded to M.A.R. originated from grants from the European Joint Program on Rare Diseases (MYOCITY), the Swiss National Science Foundation (grants 189248 and 220244) and from the cantons of Basel-Stadt and Basel-Landschaft. J.D. received support from the National Science Centre MAESTRO grant (2018/30/A/NZ3/00412). J.S. received support from the Priority Research Area BioS under the Strategic Program Excellence Initiative at Jagiellonian University.

## Author contributions

Conceptualization: T.J.M., J.D., and M.A.R.; methodology: T.J.M.; investigations: T.J.M., N.L., J.R.R., M.B., S.L., J.S., and K.S.; statistical analysis: T.J.M.; visualization: T.J.M.; supervision: M.A.R. and J.D.; writing original draft: T.J.M. and M.A.R.; writing review and editing: T.J.M. and M.A.R.; funding acquisition: M.A.R. and J.D.

## Competing interests

The authors declare no competing interests.
