## [Peer Review File · Nature Communications]

Loss of cell-autonomously secreted laminin- α 2 drives muscle stem cell dysfunction in LAMA2-related muscular dystrophy

Corresponding Author: Professor Markus Rüegg

Version 0:

Reviewer comments:

Reviewer #1

(Remarks to the Author)

This paper shows convincingly that muscle stem cells produce laminin A2 that is incorporated into their micro environment. Without this, the stem cells show reduced proliferation whilst differentiation is unimpaired. This reduced proliferation delays regeneration, with catch up occurring in the same time frame as the restoration of laminin A2 levels (presumably by other cell types). The cell autonomous nature of the defect is shown by both transplantation and cell type specific conditional knockout strategies, and studies with human iPS derived myoblasts confirm direct relevance to human congenital muscular dystrophy.

The role of laminin A2 in stem cell behaviour is important and interesting. The experimental strategy is logical, thorough and ambitious and the results are clear. With the use of transplantation, conditional knockouts and human studies I do not think further experimental work is required for publication. I have only a few very minor comments

Line 122. Has the low level of laminin A2 in quiescent stem cells being shown in vivo in figures 1A, 1B? Also, does Figure 1 E convincingly show low levels on the few quiescent stem cells analysed?

Line 223. To a non muscle biologist the use of MuSC and then PM to describe what appears to be the same preparation is confusing.

Line 592 and beyond. Another mechanism by which laminin A2 might regulate proliferation is via growth factor signalling amplification and/or switching. See Nature Cell Biology 4, 833 (2002)

In figure 5 would it be possible to show the recovery of laminin A2 levels on day 10 by IF as well as by western blot? It's an interesting point, and I didn't find the western blot that convincing

Reviewer #2

(Remarks to the Author)

In this article, the authors demonstrate that laminin- α 2 is secreted by muscle stem cells (MuSCs) and that this secretion is impaired under pathological conditions, particularly in the context of LAMA2-related muscular dystrophy, contributing to MuSC dysfunction.

The authors present convincing data showing laminin- α 2 expression in MuSCs. They then use multiple complementary models. including DyW and Dy3K mice, which are established models of laminopathies, a conditional MuSC-specific laminin- α 2 knockout, and human iPSC-derived MuSCs, to show that loss of laminin- α 2 leads to reduced MuSC proliferation in vitro, ex vivo, and in vivo. Their results are robust and strongly support the conclusion that laminin- α 2 loss contributes to cell-intrinsic MuSC impairment in LAMA2 muscular dystrophy. These findings are a novel contribution to the emerging field of satellite cell-opathies.

However, some additional experiments would strengthen the authors' claims:

Major points :

- 1) The authors show convincing evidence that Lama2 is expressed in MuSCs. What about the expression of the other isoforms of laminins (both at the RNA and protein levels)? Are the MuSCs capable of expressing all the other chains needed to form laminin 211. If these other isoforms are expressed, how are they affected in Lama2-KO?
- 2) A major limitation of the current study is the absence of a mechanistic link between laminin- α 2 loss and the observed proliferation defects. Specifically: What are the signaling pathways downstream of laminin- α 2 that regulate proliferation? Are known ECM receptors involved? The authors have added a brief paragraph in the discussion addressing potential mechanisms, but for a high-impact paper, I believe this should be supported by experimental data.
- 3) In Figures 1h and 5n, the authors use immunofluorescence on injured muscle cross-sections to localize laminin- α 2 in myogenic cells. While the images are of high quality, the strong expression surrounding myofibers makes it difficult to clearly distinguish its specific localization in MuSCs. A Z-stack 3D reconstruction would help better define this expression. Additionally, a more precise quantification of Lama2 expression specifically in MuSCs (e.g., on isolated single myofibers Fig. 5d) would strengthen the authors' conclusions.
- 4) In Figures 1i and 5p, the authors conclude that laminin- α 2 protein levels increase at 4 DPI. However, based on the presented blots, this upregulation is not clearly convincing. Full blots, including Ponceau staining, should be included in supplemental figures. Moreover, panel 1j lacks a statistical test, which suggests that the difference between uninjured and 4 DPI may not be significant. Better representative images and quantitative analysis with proper normalization and statistical validation is needed to support this conclusion.
- 5) In their single myofibers experiments, did the authors look at Pax7 and MyoD staining to determine if lack of Lama2 could also play a role in the regulation of self-renewal (in addition to regulation cell cycle).
- 6) The authors show in vitro that laminin α 2 KO MuSCs display proliferation defects. Adding exogenous laminin- α 2 in vitro to determine whether MuSC proliferation defects can be rescued would be useful to validate the cell-intrinsic defect.
- 7) Can the authors assess specifically expression of cell-cycle regulators to consolidate how proliferation is dysregulated.
- 8) It would be helpful to know at least at the transcriptomic level whether other matrix components are altered upon laminin- α 2 loss, both in MuSCs and in the surrounding tissue.

Minor points:

- Line 439-440: it is mentioned: "we observed that TAs from control mice contained a higher proportion of fully-fused EGFP+ fibers than TAs from MuSC-Lama2KO mice at 4 DPI". It is unclear how "fully-fused" fibers are quantified.
- The authors used the contralateral muscle for many experiments. Would it be possible that MuSCs that are in Galert in the contralateral muscle produce Lama2 to prepare for activation?

Reviewer #3

(Remarks to the Author)

Version 1:

Reviewer comments:

Reviewer #1

(Remarks to the Author)

This revised manuscript has addressed the reviewer's comments well and is improved as a consequence. I find it, as I said in my first review, convincing and important.

Minor comments

I think that including the new data on A2 expression in quiescent and activated (proliferating) cells in the supplementary material would be useful. Future work will examine the relative contributions of inside out and outside in signalling to integrin activation and downstream pathway activation, and this data would be helpful for that work.

The addition of the data on the expression of other laminins is valuable

The work on integrin activation is interesting, but complicated by the likely inclusion of other non-laminin binding beta 1 containing integrins (I deduce that 9EG7 has been used, which also sees activated α 5 β 1). Nonetheless I think it could be included (along with a more explicit guide as to the antibody used), but rather than just say "trend" why not include within the text the p value shown in Fig S7 (0.0512) so the readers can decide for themselves as to the value of the experiment.

Reviewer #2

(Remarks to the Author)

In this revised version, the authors have addressed many of the concerns previously raised. Some aspects, such as the underlying mechanism, could have been developed further; however, I acknowledge that this would likely constitute a separate project beyond the scope of the current article.

While I recommend acceptance of the manuscript, there is one issue that remains insufficiently addressed: the Western blots of Lama2 in Figures 1i–j and 5p–q. As the authors themselves noted, Lama2 detection by Western blot is technically challenging, and the samples display substantial variability. I remain unconvinced by the data presented. Although the manuscript's wording has been toned down, the current presentation could still be misleading.

Ideally, these data should be re-analyzed with additional samples to account for the variability. If this is not feasible, I recommend either removing these data entirely or minimally moving them to the supplementary material and further softening the text, limiting the statement to noting that variability precludes a clear interpretation. I do not consider these data essential to the manuscript, and it is preferable to present incomplete data than to risk misleading conclusions or overinterpretation.

That being said, I would like to congratulate the authors on a very interesting and valuable contribution to the field.

Reviewer #3

(Remarks to the Author)

Point-by-point response to the reviewers' comments

REVIEWER COMMENTS

Reviewer #1 (Remarks to the Author):

This paper shows convincingly that muscle stem cells produce laminin A2 that is incorporated into their micro environment. Without this, the stem cells show reduced proliferation whilst differentiation is unimpaired. This reduced proliferation delays regeneration, with catch up occurring in the same time frame as the restoration of laminin A2 levels (presumably by other cell types). The cell autonomous nature of the defect is shown by both transplantation and cell type specific conditional knockout strategies, and studies with human iPS derived myoblasts confirm direct relevance to human congenital muscular dystrophy.

The role of laminin A2 in stem cell behaviour is important and interesting. The experimental strategy is logical, thorough and ambitious and the results are clear. With the use of transplantation, conditional knockouts and human studies I do not think further experimental work is required for publication. I have only a few very minor comments

We thank the reviewer for his/her positive comments and enthusiasm towards our research.

Line 122. Has the low level of laminin A2 in quiescent stem cells being shown *in vivo* in figures 1A, 1B? Also, does Figure 1 E convincingly show low levels on the few quiescent stem cells analysed?

We thank the reviewer for raising this point. Figures 1a, 1b, 1c and 1d show that primary myoblasts (i.e., cultured MuSCs) that are *Pax7*-positive express *Lama2* and the laminin- $\alpha 2$ protein while they lose transcript and protein expression when they start differentiating (*Myog/Myogenin*-positive). The same pattern is seen *in vivo*, as shown by single-cell RNA sequencing (Fig. 1e). These data were generated using a published dataset¹.

As the reviewer seemed skeptical about the De Micheli data¹, we tried to confirm these data by combining single molecule RNA fluorescence *in situ* hybridization (smRNA FISH) with immunostaining *in vivo*. In uninjured muscle (N=7 mice), we detected *Lama2* mRNA in approximately 97% of quiescent Ki67-/Pax7+ cells (Reviewer Fig. 1a, 1b). We confirmed this *ex vivo* with freshly-isolated primary myoblasts (PMs) where 95% of *Pax7*-expressing cells expressed *Lama2* (Reviewer Fig. 1c, 1d). After the first passage (at approximately 3 days post plating), this number increased to 100% of *Pax7*-expressing cells (Reviewer Fig. 1c, 1d). As smRNA FISH is a quantitative method, we also counted the number of dots per cell. In freshly-isolated PMs, an average of 4 *Lama2* mRNAs per cell was counted, whilst after the first passage, cells had approximately 14 mRNAs per cell (Reviewer Fig. 1c, 1e). These quantifications indicate that most quiescent MuSCs express *Lama2* and that, as the cells activate and proliferate *ex vivo*, *Lama2* expression increases. Thus, our data are in line with the data shown in Figures 1a - e.

We have decided not to include the new data in the paper but we are happy to do so if wished by the reviewer and/or the editor.

Reviewer Fig. 1: *Lama2* is expressed by quiescent and activated MuSCs. a

Representative smRNA FISH combined with immunostaining in uninjured *tibialis anterior* muscle (*Lama2* in magenta, Pax7 in green, Ki67 in orange, DAPI in blue). Green arrows indicate Ki67-/Pax7+ quiescent MuSCs. **b** Quantification of the proportion of Ki67-/Pax7+ quiescent MuSCs expressing *Lama2* (n= 7 wild-type mice). **c** Representative smRNA FISH in primary myoblasts immediately post-sorting or after the first passage (*Pax7* in green, *Lama2* in magenta, DAPI in blue). **d** Quantification of the proportion of *Pax7*-expressing cells that express *Lama2* immediately post-sorting and after the first passage. **e** Quantification of the number of *Lama2* mRNAs per *Pax7*-expressing cell immediately post-sorting and after the first passage. In **e**, statistical significance was determined by paired student's two-sided t-test. **** $P < 0.0001$.

Line 223. To a non muscle biologist the use of MuSC and then PM to describe what appears to be the same preparation is confusing.

“Primary Myoblast” (PMs) is a term that refers to MuSCs that are cultured *in vitro*. The reason for this nomenclature is the fact that MuSCs in culture lose quiescence, spontaneously activate and proliferate, and have a reduced regenerative capacity compared to freshly-isolated MuSCs after transplantation^{2,3,4}. We now define this term more clearly in the revised manuscript (line 95) and use it consistently.

Line 592 and beyond. Another mechanism by which laminin A2 might regulate proliferation is via growth factor signalling amplification and/or switching. See Nature Cell Biology 4, 833 (2002)

We thank the reviewer for sharing this insight. We have now added this hypothesis to the Discussion (line 630):

“Additionally, laminin- α 2 could rely on specific growth factors to exert its mitogenic effect. This hypothesis is supported by the observation that in the central nervous system, laminin- α 2 binds to receptors on oligodendrocytes and influences the cells’ fate by amplifying and/or switching their response to neuregulin-1⁵⁹. A similar mechanism, relying on growth factors that are present in the niche of activated MuSCs, could explain why laminin- α 2 does not promote the proliferation of quiescent MuSCs.”

In figure 5 would it be possible to show the recovery of laminin A2 levels on day 10 by IF as well as by western blot? It's an interesting point, and I didn't find the western blot that convincing

We have now added representative images (Fig. S5m) and the quantification (Fig. S5n) of laminin- α 2 staining intensity in muscle cross-sections at 10 DPI. Overall, the Western blot and immunostaining data show that there is a lower abundance laminin- α 2 in MuSC-*Lama2*KO mice at 4 DPI, but not at 10 DPI. This suggests that other cell types may eventually compensate for the loss of laminin- α 2 secretion.

Reviewer #2 (Remarks to the Author):

In this article, the authors demonstrate that laminin- α 2 is secreted by muscle stem cells (MuSCs) and that this secretion is impaired under pathological conditions, particularly in the context of LAMA2-related muscular dystrophy, contributing to MuSC dysfunction. The authors present convincing data showing laminin- α 2 expression in MuSCs. They then use multiple complementary models, including DyW and Dy3K mice, which are established models of laminopathies, a conditional MuSC-specific laminin- α 2 knockout, and human iPSC-derived MuSCs, to show that loss of laminin- α 2 leads to reduced MuSC proliferation *in vitro*, *ex vivo*, and *in vivo*. Their results are robust and strongly support the conclusion that laminin- α 2 loss contributes to cell-intrinsic MuSC impairment in LAMA2 muscular dystrophy. These findings are a novel contribution to the emerging field of satellite cellopathies.

We thank the reviewer for his/her positive comments.

However, some additional experiments would strengthen the authors' claims:

Major points :

1) The authors show convincing evidence that *Lama2* is expressed in MuSCs. What about the expression of the other isoforms of laminins (both at the RNA and protein levels)? Are the MuSCs capable of expressing all the other chains needed to form laminin 211. If these other isoforms are expressed, how are they affected in *Lama2*-KO?

We thank the reviewer for raising this important point. We have now re-analyzed single-cell RNA sequencing datasets from De Micheli et. al¹ and show that *Lamb1* and *Lamc1*, encoding laminin- β 1 and laminin- γ 1, respectively, are also expressed by MuSCs (New Fig. S1g). We also examined the expression of laminin proteins in cultures of single fibers. In this

preparation, as MuSCs activate and proliferate, they secrete laminin- β 1- γ 1 concomitantly with laminin- α 2 (New Fig. S1h). We included these results in the manuscript (line 157, New Fig. S1g, S1h). Amongst the other laminin- β and - γ subunits, *Lamb3*, *Lamc2* and *Lamc3* show little to no expression in MuSCs (Reviewer Fig. 2). *Lamb2* is also expressed in MuSCs (Reviewer Fig. 2).

Reviewer Fig. 2: Expression of *Lamb* and *Lamc* genes in MuSCs. Violin plots showing the expression of *Lamb1*, *Lamb2*, *Lamb3*, *Lamc1*, *Lamc2* and *Lamc3* in a published single-cell RNA sequencing dataset¹. These genes encode laminin- β 1, - β 2, - β 3, - γ 1, - γ 2 and - γ 3 subunits, respectively.

Regarding the other laminin- α chains, it is known that the basement membrane surrounding the muscle fibers of LAMA2 MD patients and mouse models thereof contains high levels of laminin- α 4 as a “compensatory” laminin- α chain^{5, 6, 7}. As these data do not address the cell origin of laminin- α 4, we now examined laminin- α 4 secretion in MuSCs. To this end, we stained injured wild-type, *dy^W/dy^W*, control, and MuSC-*Lama2*KO muscles with an anti-laminin- α 4 antibody. In wild-type, control and MuSC-*Lama2*KO mice, laminin- α 4 was present in puncta, indicative of blood vessels (magenta arrows), while in *dy^W/dy^W* mice it also surrounded muscle fibers (green arrows) (Reviewer Fig. 3a) and consequently covered a higher proportion of the muscle (Reviewer Fig. 3b). This indicates that the compensatory upregulation of laminin- α 4 only occurred in *dy^W/dy^W* mice.

We also tested whether this compensatory upregulation was due to laminin- α 4 secretion by *dy^W/dy^W* MuSCs, by performing smRNA FISH on wild-type and *dy^W/dy^W* regenerating *tibialis anterior* muscles at 4 DPI. Consistent with the immunohistochemistry data above, the number of *Lama4*-positive puncta was much higher in *dy^W/dy^W* muscle than in wild-type mice (Reviewer Fig. 3c), but these puncta did not co-localize with *Pax7*-expressing cells (Reviewer Fig. 3c, white arrows). Similarly, they did not overlap with *Myog*-positive regions (Reviewer Fig. 3c). Hence, *Lama4* expression is not altered in *Pax7*- or *Myog*-positive MuSCs in *dy^W/dy^W* mice.

Finally, we analyzed recently generated single-nucleus RNA sequencing data from uninjured wild-type and *dy^W/dy^W* muscle. As shown in Reviewer Fig. 3d, we did not detect a significant increase in *Lama4* expression in *dy^W/dy^W* MuSCs (unpublished data). While

Lama2 in $dy^{W/dy^{W}}$ MuSCs is low – consistent with our finding that wild-type MuSCs express *Lama2* – the only transcript that is slightly increased in MuSCs is *Lama5* (Reviewer Fig. 3d). This is consistent with a transient expression of laminin- $\alpha5$ during muscle regeneration^{8, 9}. Finally, we stained laminin- $\alpha4$ in cell culture dishes and detected very little to no signal for all samples (Reviewer Fig. 3e).

In conclusion, these experiments demonstrate that the loss of laminin- $\alpha2$ production by MuSCs does not lead to a compensatory upregulation of laminin- $\alpha4$ in those cells, suggesting that the laminin- $\alpha4$ present in $dy^{W/dy^{W}}$ muscle likely comes from other cell types.

Reviewer Fig. 3: *Lama2*-deficient MuSCs do not upregulate laminin- $\alpha4$ production.

a Representative immunostaining of wild-type and $dy^{W/dy^{W}}$ TAs at 14 DPI, and control and MuSC-*Lama2*KO TAs at 10 DPI (laminin- $\alpha4$ in white, DAPI in blue). In the lower panels, magenta arrows point to the punctate staining of blood vessels and green arrows show laminin- $\alpha4$ surrounding muscle fibers. **b** Quantification of the proportion of tissue stained for

laminin- α 4. **c** Single molecule RNA FISH in wild-type and dy^{W}/dy^{W} TAs at 4 DPI (*Pax7* in green, *Myog* in orange, *Lama4* in magenta, DAPI in blue). White arrows indicate *Pax7*-expressing cells. **d** Violin plot showing the expression of laminin isoforms in dy^{W}/dy^{W} MuSCs compared to wild-type MuSCs in uninjured *quadriceps* muscle of 4-week-old mice. This single-nucleus RNA sequencing data is unpublished (manuscript in preparation). **e** Representative immunostaining of wild-type, dy^{W}/dy^{W} , control and MuSC-*Lama2*KO PMs cultured in proliferation medium for 5 days on collagen-coated plates (laminin- α 4 in white, DAPI in blue). Data are means \pm SEM. In **b**, statistical significance was determined by unpaired student's two-sided t-test. ** $P < 0.01$.

2) A major limitation of the current study is the absence of a mechanistic link between laminin- α 2 loss and the observed proliferation defects. Specifically: What are the signaling pathways downstream of laminin- α 2 that regulate proliferation? Are known ECM receptors involved? The authors have added a brief paragraph in the discussion addressing potential mechanisms, but for a high-impact paper, I believe this should be supported by experimental data.

We agree that identifying a possible mechanistic link between the loss of laminin- α 2 and the proliferative defects would strengthen our findings. As we discuss in the manuscript (line 644), one hypothesis is that self-secreted laminin- α 2 supports MuSC proliferation by binding to receptors on the cell and activating mitogenic signaling pathways. To investigate this, we have now assessed phosphorylation of Erk1/2 and Akt, two downstream signaling effectors of ECM-binding receptors in MuSCs¹⁰. In cell culture, we observed that in dy^{W}/dy^{W} PMs phosphorylation of Erk1/2 was significantly reduced compared to wild-type PMs, but no significant changes in Akt phosphorylation were seen (New Fig. S7a, S7b). In MuSC-*Lama2*KO PMs, phosphorylation of both Erk1/2 and Akt were slightly lower compared to control PMs (New Fig. S7a, S7b). This indicates the presence of common and distinct pathways that affect proliferation of *Lama2*-deficient cells.

Since β 1-integrin-containing receptors are known laminin-211 receptors¹¹ and MuSC-specific β 1-integrin knockout causes a reduction in proliferation concomitant with a downregulation of Erk1/2 and Akt phosphorylation¹⁰, we next assessed the activation state of β 1-integrin in wild-type, dy^{W}/dy^{W} , control and MuSC-*Lama2*KO PMs by staining the cells using an antibody that recognizes β 1-integrin in its active conformation^{12, 13}. Quantification of the staining intensity shows a trend towards less activated β 1-integrin on dy^{W}/dy^{W} PMs (New Fig. S7c), though the difference did not reach statistical significance (New Fig. S7d). MuSC-*Lama2*KO PMs did not show lower levels of β 1-integrin activation compared to control cells (New Fig. S7c, S7d). Since *Itgb1* expression levels were comparable between samples (New Fig. S7e), these results indicate that lower activation of β 1-integrin-containing receptors may contribute to the slower proliferation of dy^{W}/dy^{W} MuSCs. However, the results also suggest that additional β 1-integrin-independent mechanisms are at play. These could include binding to the dystroglycan receptor complex, which is the other important laminin- α 2 receptor in muscle, or receptor-independent mechanisms, which we discuss in the manuscript (line 655). We include all the new results in the manuscript's Results section (line 481, New Fig. S7) and we discuss their implications in the Discussion (line 649).

3) In Figures 1h and 5n, the authors use immunofluorescence on injured muscle cross-sections to localize laminin- α 2 in myogenic cells. While the images are of high quality, the strong expression surrounding myofibers makes it difficult to clearly distinguish its specific localization in MuSCs. A Z-stack 3D reconstruction would help better define this expression. Additionally, a more precise quantification of *Lama2* expression specifically in MuSCs (e.g., on isolated single myofibers Fig. 5d) would strengthen the authors' conclusions.

In the manuscript, we describe the presence of laminin- α 2 on the basal and apical side of MuSCs post-injury (line 128 – 135). Due to the resolution of the images, we had not determined whether laminin- α 2 was located within MuSCs. We have now included a 3D reconstruction showing laminin- α 2 inside and around MuSCs at 4 DPI. In this movie, it is also

apparent that laminin- α 2 surrounds MuSCs located within ghost fibers. This video is included with the manuscript as New Supplementary Movie 1.

To quantify *Lama2* expression in MuSCs, we determined the proportion of MuSCs expressing *Lama2* *in vivo* and *ex vivo* using single molecule RNA FISH (Reviewer Fig. 1a – d). These experiments demonstrated that over 95% of wild-type MuSCs express *Lama2* *in vivo* and *ex vivo* (Reviewer Fig. 1a – d).

4) In Figures 1i and 5p, the authors conclude that laminin- α 2 protein levels increase at 4 DPI. However, based on the presented blots, this upregulation is not clearly convincing. Full blots, including Ponceau staining, should be included in supplemental figures. Moreover, panel 1j lacks a statistical test, which suggests that the difference between uninjured and 4 DPI may not be significant. Better representative images and quantitative analysis with proper normalization and statistical validation is needed to support this conclusion.

Indeed, laminin- α 2 detection by Western blot is challenging because of its size and the rather high variability between mice. We now include the full Ponceau stains and blots in Reviewer Fig. 4. The same data are part of “source data” of the manuscript. We also added a negative control (a *tibialis anterior* sample from *dy^{3K}/dy^{3K}* mice, a LAMA2 MD mouse model) so that readers see the size of laminin- α 2 and the specificity of the anti-laminin- α 2 antibody (Reviewer Fig. 4, lower panel).

We have now also adapted Fig. 1j according to the reviewer’s suggestion. In this adapted presentation, one can appreciate the strong variation in laminin- α 2 abundance between uninjured samples at 4 DPI. This is the reason we had initially normalized the abundance of laminin- α 2 in the injured TA to that in the uninjured (contralateral) TA of the same mouse. We now use lines to connect injured and uninjured TAs of the same mouse. Although the high variation prevents that the mean levels of laminin- α 2 in uninjured and injured TAs at 4 and 10 DPI become significantly different, we hope that this new presentation allows the reader to appreciate that the amount of laminin- α 2 increases in the injured TA at 4 DPI in each mouse and that the amount is lower at 10 DPI. We have adapted the results section (line 136) and the Fig. 1j legend (line 1433) accordingly.

Reviewer Fig. 4: Full blots and Ponceau stains relating to Figures 1i and 5p. For each experiment, the Ponceau stain is shown on the left, and the uncropped blot on the right. On the blot, the box with dashed line represents the cropped area shown in the manuscript. Laminin- α 2 has a predicted molecular mass of 390 kDa but is detected at lower molecular masses because of proteolytic cleavage¹⁴. The blot comparing control and MuSC-*Lama2*KO TAs at 10 DPI (lowest panel) contains proteins extracted from a *dy*^{3K}/*dy*^{3K} TA as a negative control. This sample does not display a band between 268 and 460 kDa. These data are included in the source data file of the manuscript.

5) In their single myofibers experiments, did the authors look at Pax7 and MyoD staining to determine if lack of Lama2 could also play a role in the regulation of self-renewal (in addition to regulation cell cycle).

Thank you for this question. We performed another single fiber experiment with EDLs from control and MuSC-*Lama2*KO mice and determined the proportion of MyoD+/EGFP+ cells after 72 hours in culture (T72). At this time point, the number of MyoD+ cells was not different between the samples (New Fig. S5f; Reviewer Fig. 5). Since there were also no significant differences in the proportion of Pax7+ and Myogenin+ EGFP+ cells at T72, we conclude that the loss of laminin- α 2 in MuSCs does not influence their capacity to self-renew or differentiate on single fibers. We have included this new data in the manuscript (New Fig. S5f; discussed in line 425).

Reviewer Fig. 5: MuSC-specific *Lama2* knockout does not influence self-renewal or differentiation. **a** Representative immunostaining of single EDL fibers after 72 hours in culture (T72) (EGFP in green, MyoD in magenta, DAPI in blue). **b** Quantification of the proportion of Pax7+, MyoD+ and Myogenin+ EGFP+ cells at T72.

6) The authors show in vitro that laminin α 2 KO MuSCs display proliferation defects. Adding exogenous laminin- α 2 in vitro to determine whether MuSC proliferation defects can be rescued would be useful to validate the cell-intrinsic defect.

Again, we thank the reviewer for this suggestion. In fact, we had discussed such experiments during the project. Unfortunately, laminin-211 (consisting of α 2, β 1 and γ 1 chains) is proteolytically cleaved¹⁴. For example, it is well known that an 80 kD-size fragment is generated. Hence, the resulting α 2 chain is smaller than the full-length size of 390 kDa and this is also the case for the laminin- α 2 chain of the commercially-sold laminin-211. This makes it difficult to interpret the results and we decided not to test laminin-211. Instead, we tested whether the proliferative defects of *Lama2*-deficient PMs observed on collagen-coated plates (Fig. S6e) persisted on MatrigelTM-coated plates, as MatrigelTM predominantly contains laminin-111, which is not proteolytically cleaved¹⁵ and has been shown to restore muscle function when expressed in *Lama2* knockout mice¹⁶. MatrigelTM did not rescue the proliferation of *Lama2*-deficient PMs (Reviewer Fig. 6). This is evidence that coating the cell

culture dishes does not recapitulate the localization and abundance of self-secreted laminin- $\alpha 2$. We hypothesize that this might be the reason why the transplantation of dy^{W}/dy^{W} MuSCs (Fig. 4) and the inducible MuSC-specific *Lama2* knockout (Fig. 5), which assessed the proliferation of *Lama2*-deficient MuSCs in an environment that contains laminin- $\alpha 2$, did not rescue their proliferation.

Reviewer Fig. 6: Culturing *Lama2*-deficient PMs on laminin-containing Matrigel™ does not rescue their proliferation deficit. Quantification of the proportion of EdU+/EGFP+ MuSCs after a 15h incubation with EdU on collagen or Matrigel™-coated plates.

7) Can the authors assess specifically expression of cell-cycle regulators to consolidate how proliferation is dysregulated.

Please see our experiments described in the answer to your comment #2 about the mechanisms involved. Erk1/2 and Akt are key regulators of mitogenic signaling pathways and show lower activation in *Lama2*-deficient cells (see New Fig. S7a, S7b). These data reinforce our finding that *Lama2*-deficient MuSCs have impaired proliferation. We have included these results in the manuscript (line 481, New Fig. S7a, S7b).

8) It would be helpful to know at least at the transcriptomic level whether other matrix components are altered upon laminin- $\alpha 2$ loss, both in MuSCs and in the surrounding tissue.

In response to this remark, we conducted the following experiments:

1. We have assessed whether the loss of laminin- $\alpha 2$ production by MuSCs leads to an increase in the deposition of collagen post-injury by Sirius Red staining. There were no differences in the proportion of tissue stained by Sirius Red (Fig. S5f).
2. We have determined that while laminin- $\alpha 4$ abundance is increased in dy^{W}/dy^{W} muscle post-injury, *Lama2*-deficient MuSCs do not upregulate *Lama4* expression or laminin- $\alpha 4$ secretion (Reviewer Fig. 3a – f, see response to major comment #1 for more detail).
3. Since fibro-adipogenic progenitor cells (FAPs) are known secretors of ECM, we compared the transcriptome of wild-type and dy^{W}/dy^{W} FAPs in a single-nucleus RNA sequencing dataset (unpublished data; manuscript in preparation). Indeed, numerous ECM genes, including those encoding collagen and periostin, were differentially expressed in dy^{W}/dy^{W} FAPs (Reviewer Fig. 7). However, not all genes were upregulated; many were in fact downregulated (Reviewer Fig. 7).

Reviewer Fig. 7: Numerous genes are differentially expressed in *dy^w/dy^w* FAPs.

Volcano plot showing the 200 most differentially expressed genes in *dy^w/dy^w* FAPs compared to wild-type FAPs (unpublished single-nucleus RNA sequencing). Only the names of genes encoding ECM proteins are shown.

Minor points:

- Line 439-440: it is mentioned: “we observed that TAs from control mice contained a higher proportion of fully-fused EGFP+ fibers than TAs from MuSC-Lama2KO mice at 4 DPI”. It is unclear how “fully-fused” fibers are quantified.

We qualified “fully-fused” fibers as fibers that were surrounded by a basement membrane and homogenously filled by EGFP. We have clarified the quantification of fully-fused fibers in the legend of Fig. 5n (line 1555).

- The authors used the contralateral muscle for many experiments. Would it be possible that MuSCs that are in Galert in the contralateral muscle produce Lama2 to prepare for activation?

Good point again. Of course, what you write is possible. However, since laminin- α 2 abundance varies so much from mouse to mouse (Fig. 1i, 5p), we decided to use this method in Fig. 5q to normalize quantifications of laminin- α 2 abundance in the injured TA to the contralateral TA of the same mouse. For measurements of injured muscle mass (Fig. 2b, 5h, S3a) and quantifications of the number of fibers per cross-section (Fig. 2c, 5m) normalization to the contralateral uninjured muscle takes into account this baseline variability between mice.

Even if this method may have pitfalls, our single-cell RNA sequencing (Fig. 1e) and smRNA FISH (Reviewer Fig. 1a, 1b) data show that *Lama2* is expressed in quiescent MuSCs and that its expression increases as the cells activate and proliferate *ex vivo*

(Reviewer Fig. 1c, 1e). We have not specifically compared *Lama2* expression in quiescent and Galert MuSCs.

Reviewer #3 (Remarks to the Author):

We thank the reviewer for his/her contribution.

References

1. De Micheli AJ, *et al.* Single-Cell Analysis of the Muscle Stem Cell Hierarchy Identifies Heterotypic Communication Signals Involved in Skeletal Muscle Regeneration. *Cell Rep* **30**, 3583-3595 e3585 (2020).
2. Collins CA, *et al.* Stem cell function, self-renewal, and behavioral heterogeneity of cells from the adult muscle satellite cell niche. *Cell* **122**, 289-301 (2005).
3. Montarras D, *et al.* Direct isolation of satellite cells for skeletal muscle regeneration. *Science* **309**, 2064-2067 (2005).
4. Sacco A, Doyonnas R, Kraft P, Vitorovic S, Blau HM. Self-renewal and expansion of single transplanted muscle stem cells. *Nature* **456**, 502-506 (2008).
5. Reinhard JR, *et al.* Linker proteins restore basement membrane and correct LAMA2-related muscular dystrophy in mice. *Sci Transl Med* **9**, (2017).
6. Reinhard JR, Porrello E, Lin S, Pelczar P, Previtali SC, Rugg MA. Nerve pathology is prevented by linker proteins in mouse models for LAMA2-related muscular dystrophy. *PNAS Nexus* **2**, pgad083 (2023).
7. Azimi M, Niimi T, Yoshida N, Kitagawa Y. Differential expression of mRNAs encoding laminin chain variants during in vitro development of mouse blastocysts. *Cytotechnology* **31**, 185-193 (1999).
8. Rayagiri SS, *et al.* Basal lamina remodeling at the skeletal muscle stem cell niche mediates stem cell self-renewal. *Nature Communications* **9**, (2018).
9. Sorokin LM, *et al.* Laminin alpha4 and integrin alpha6 are upregulated in regenerating dy/dy skeletal muscle: comparative expression of laminin and integrin isoforms in muscles regenerating after crush injury. *Exp Cell Res* **256**, 500-514 (2000).
10. Rozo M, Li L, Fan CM. Targeting beta1-integrin signaling enhances regeneration in aged and dystrophic muscle in mice. *Nat Med* **22**, 889-896 (2016).
11. Nishiuchi R, *et al.* Ligand-binding specificities of laminin-binding integrins: a comprehensive survey of laminin-integrin interactions using recombinant

- alpha3beta1, alpha6beta1, alpha7beta1 and alpha6beta4 integrins. *Matrix Biol* **25**, 189-197 (2006).
12. Bazzoni G, Shih DT, Buck CA, Hemler ME. Monoclonal antibody 9EG7 defines a novel beta 1 integrin epitope induced by soluble ligand and manganese, but inhibited by calcium. *J Biol Chem* **270**, 25570-25577 (1995).
 13. Takagi J, Springer TA. Integrin activation and structural rearrangement. *Immunol Rev* **186**, 141-163 (2002).
 14. Talts JF, Mann K, Yamada Y, Timpl R. Structural analysis and proteolytic processing of recombinant G domain of mouse laminin alpha2 chain. *FEBS Lett* **426**, 71-76 (1998).
 15. Yurchenco PD. Basement membranes: cell scaffoldings and signaling platforms. *Cold Spring Harb Perspect Biol* **3**, (2011).
 16. Gawlik K, Miyagoe-Suzuki Y, Ekblom P, Takeda S, Durbeej M. Laminin alpha1 chain reduces muscular dystrophy in laminin alpha2 chain deficient mice. *Hum Mol Genet* **13**, 1775-1784 (2004).

Point-by-point response to the reviewers' comments

REVIEWER COMMENTS

Reviewer #1 (Remarks to the Author):

This revised manuscript has addressed the reviewer's comments well and is improved as a consequence. I find it, as I said in my first review, convincing and important.

We thank the reviewer for his/her positive comments and help during the peer review process.

Minor comments

I think that including the new data on A2 expression in quiescent and activated (proliferating) cells in the supplementary material would be useful. Future work will examine the relative contributions of inside out and outside in signalling to integrin activation and downstream pathway activation, and this data would be helpful for that work.

We have now included the data comparing *Lama2* expression in freshly-isolated and proliferating primary myoblasts *ex vivo* (Fig. S1b, S1c, line 101) and the data demonstrating that quiescent MuSCs express *Lama2 in vivo* (Fig. S1f, S1g, line 120).

The addition of the data on the expression of other laminins is valuable

We have also included the data showing that *Lama2*-deficient MuSCs do not upregulate *Lama4* as an additional Supplementary Figure (Fig. S9, line 505).

The work on integrin activation is interesting, but complicated by the likely inclusion of other non-laminin binding beta 1 containing integrins (I deduce that 9EG7 has been used, which also sees activated $\alpha 5\beta 1$). Nonetheless I think it could be included (along with a more explicit guide as to the antibody used), but rather than just say "trend" why not include within the text the p value shown in Fig S7 (0.0512) so the readers can decide for themselves as to the value of the experiment.

We thank the reviewer for this insightful input.

We have now detailed the name of the antibody in the manuscript (lines 498, 674; Fig. S8d legend), provided a reference¹ (line 498) and completed the discussion to address this activation of $\beta 1$ -integrin-containing receptors by non-laminin proteins (line 676):

“Activation of $\beta 1$ -integrin-containing receptors on MuSCs in the absence of laminin- $\alpha 2$ is likely based on their engagement with other ligands, such as fibronectin which activates integrin $\alpha 5\beta 1$ ”

Reviewer #2 (Remarks to the Author):

In this revised version, the authors have addressed many of the concerns previously raised. Some aspects, such as the underlying mechanism, could have been developed further; however, I acknowledge that this would likely constitute a separate project beyond the scope of the current article.

While I recommend acceptance of the manuscript, there is one issue that remains

insufficiently addressed: the Western blots of Lama2 in Figures 1i–j and 5p–q. As the authors themselves noted, Lama2 detection by Western blot is technically challenging, and the samples display substantial variability. I remain unconvinced by the data presented. Although the manuscript's wording has been toned down, the current presentation could still be misleading.

Ideally, these data should be re-analyzed with additional samples to account for the variability. If this is not feasible, I recommend either removing these data entirely or minimally moving them to the supplementary material and further softening the text, limiting the statement to noting that variability precludes a clear interpretation. I do not consider these data essential to the manuscript, and it is preferable to present incomplete data than to risk misleading conclusions or overinterpretation.

We agree with the reviewer's concerns and have consequently moved the Western blots and their analyses to the Supplementary Figures:

- Old Fig. 1i and 1j → New Fig. S2a and S2b
- Old Fig. 5p and 5q → New Fig. S6l and S6m

We have also softened the text according to the reviewer's recommendation:

- Line 147: "Although variability among samples limited definitive conclusions, each mouse showed a trend towards higher levels of laminin- α 2 in the injured TA compared to the uninjured TA at 4 DPI, but not at 10 DPI (Fig. S2a, S2b)."
- Line 461: "As reported above (Fig. S2b), laminin- α 2 levels in control mice tended to be increased in injured TAs at 4 DPI (Fig. S6l, S6m)."

That being said, I would like to congratulate the authors on a very interesting and valuable contribution to the field.

We thank the reviewer for his/her positive comment and help throughout the peer review process.

Reviewer #3 (Remarks to the Author):

We thank the reviewer for his/her contribution to reviewing this manuscript.

References

1. Bazzoni G, Shih DT, Buck CA, Hemler ME. Monoclonal antibody 9EG7 defines a novel beta 1 integrin epitope induced by soluble ligand and manganese, but inhibited by calcium. *J Biol Chem* **270**, 25570–25577 (1995).